# A Machine-learning Based Marine Atmosphere Boundary Layer (MABL) Moisture Profile Retrieval Product from GNSS-RO Deep Refraction Signals

Jie Gong[1,*], Dong L. Wu[1], Michelle Badalov[2], Manisha Ganeshan[3,1], and Minghua Zheng[4]

[1]Climate and Radiation Lab, NASA Goddard Space Flight Center, Greenbelt, MD, USA
[2]Dept. of Computer Science, Univ. of Maryland, College Park, MD, USA
[3]GESTAR-II/Morgan State University, Baltimore, MD, USA
[4]University of California at San Diego, La Jolla, CA, USA

**Correspondence:** Jie Gong (Jie.Gong@nasa.gov)

**Abstract.** Marine Atmosphere Boundary layer (MABL) water vapor amount and gradient impact the global energy transport through directly affecting the sensible and latent heat exchange between the ocean and atmosphere. Yet, it is a well-known challenge for satellite remote sensing to profile MABL water vapor, especially when cloud or sharp vertical gradient of water vapor are present. Wu et al. (2022) identified good correlations between Global Navigation Satellite System (GNSS) deep refraction signal-to-noise-ratio (SNR) value and the global MABL water vapor specific humidity when the radio occultation (RO) signal is ducted by the moist planetary boundary layer (PBL), and they laid out the underlying physical mechanisms to explain such a correlation. In this work, we apply a machine-learning/artificial intelligence (ML/AI) technique to demonstrate the feasibility for profile-by-profile MABL water vapor retrieval using the SNR signal. Three convolutional neural network (CNN) models are trained using multi-months of global collocated hourly ERA-5 reanalysis and COSMIC-1, Metop-A and Metop-B 1 Hz SNR observations between 975 – 850 hPa with 25 hPa vertical resolution. The COSMIC-1 ML model is then applied to both COSMIC-1 and COSMIC-2 in other time ranges for independent retrieval and validation. Monte Carlo Dropout method was employed for the uncertainty estimation. Comparison against multiple field campaign radiosonde/dropsonde observations globally suggests SNR-ML method retrieved water vapor consistently outperforms the wetPrf/wetPf2 standard retrieval product at all six pressure levels between 975 hPa and 850 hPa, and either outperforms or achieves similar performance against ERA-5, indicating real and useful information is gained from the SNR signal albeit training was performed against the reanalysis. Climatology and diurnal cycle of MABL structure constructed from the SNR-ML technique are studied and compared to the reanalysis. Disparities of climatology suggest ERA-5 may systematically produces dry biases at high-latitudes, and wet biases in marine stratocumulus regions. The diurnal cycle amplitudes are too weak and sometimes off-phase in ERA-5, especially in Arctic and stratocumulus regions. These areas are particularly prone to PBL processes where this GNSS SNR-ML water vapor product may contribute the most.

# 1   Introduction

As a key component of Earth's lower atmosphere, planetary boundary layer (PBL) water vapor plays a pivotal role in Earth's energy budget, exerting a profound influence on weather and climate processes. It is an essential factor of the Earth's energy budget, influencing radiative forcing and consequently climate variability and long-term changes. Furthermore, PBL water vapor is instrumental in modulating local and regional weather patterns by affecting cloud formation, precipitation and temperature. Therefore, study of PBL water vapor stands as a vital element in advancing our comprehension of Earth's atmosphere and its broader implications for our planet's climate system.

70% of the Earth's surface is covered by water. The sensible and latent heat exchange between the ocean boundary and the marine atmosphere boundary layer (MABL) happens at different spatial and temporal scales, which is determined by not only ocean surface properties (e.g., wind speed, sea surface temperature) but also MABL thermodynamic structures. For example, under the context of susceptibility of polar area to the climate change, Boisvert et al. (2015) found Arctic PBL humidity and temperature biases in the reanalysis are the major error sources for the evaporation estimation compared to satellite observations. Cloud-cimate feedback is another motivation highlighted by NASA's PBL incubation study (Teixeira et al. (2021)). As another example, Milan et al. (2019) found strong correlation between MABL cloud top height and below-cloud water vapor amount using two joint satellite retrieval products.

Data sparsity is a critical problem for advancing MABL science. Satellite remote sensing undoubtedly provides the best solution in terms of global coverage, but it is very difficult to retrieve MABL WV and its vertical distribution when cloud or sea ice are present. When clouds are present in the scene, emissions from clouds often overwhelms the emission signal from the MABL water vapor and prevents passive instruments sensing the below-cloud atmosphere. When sea ice is present, scattering or surface emission from the sea ice are often inseparable from water vapor emission signals and distort the retrieval result. Taking the aforementioned two research as examples, Boisvert et al. (2015) uses Level 2 AIRS water vapor and temperature retrieval products, which are only available for clear or partially-cloudy sky situations, so it inherently contains a sampling bias. Milan et al. (2019) derived MABL total WV amount from subtracting MODIS above-cloud water vapor from AMSU-A total column water vapor, which still lacks the vertical information of WV in the MABL.

Using low-frequency microwave L-band to transmit signals along the limb path, the Global Navigation Satellite System (GNSS) satellite overcomes the two above difficulties and provides high vertical resolution (100-200 m) of the MABL water vapor under all-sky conditions. GNSS Radio Occultation (GNSS-RO) retrieves temperature and water vapor profiles using the 1D-Var approach routinely from the Level 2 bending angle product (referred as "standard L2 product" or "operational L2 product" hereafter), the latter of which is used operationally in numerical weather data assimilation systems to improve weather forecasts (e.g.,Kuo et al. (2000)). Because of the rapid growth of SmallSat/CubeSat constellations from both the commercial and non-profit sectors, GNSS-RO technique provides a promising future for the needed global spatial-temporal sampling of MABL WV and its variability. Like other limb sounders, the disadvantage of GNSS-RO is its relatively coarse horizontal resolution (several hundred kilometers) that smears out horizontally inhomogeneous signals. This is typically not a big concern in MABL as the vertical gradient is much sharper than the horizontal gradient and harder to characterize.

However, GNSS-RO WV retrieval profiles have excessively high failure rate in the MABL. That is because the GNSS-RO signal-to-noise ratio (SNR) decreases with decreasing altitude due to the atmospheric defocusing effect, and the Level-2 RO signal hence often does not meet the SNR threshold near the surface. As a result, the GNSS-RO 1D-Var based retrievals often fail in the MABL due to weak RO signals. Fig. 1 gives an example of the success statistics (%) as a function of height for temperature (Fig. 1a) and water vapor (Fig. 1b) over the tropical ocean ($10°S - 10°N$). Using $0.5km$ and $1km$ above the

ocean surface as the reference lines, we can see although the COSMIC-2 (Constellation Observing System for Meteorology, Ionosphere, and Climate-2) has significantly improved its SNR compared to its predecessor COSMIC-1, the success rate is still about 60% at $0.5km$ and slightly over 70% at $1km$ for the GNSS-RO WV retrieval, while this number is only 40% and 55% for COSMIC-1 at respective altitudes. The low SNR widely exists for commercial GNSS satellites especially in the lowest 500 m above the surface (Ganeshan et al. (2025)). Moreover, even passed the SNR threshold, some bending angle profiles are

significantly biased in the PBL when ducting happens because the refractivity index becomes negative, which leads to biases in the operational water vapor retrievals (Feng et al. (2020)).

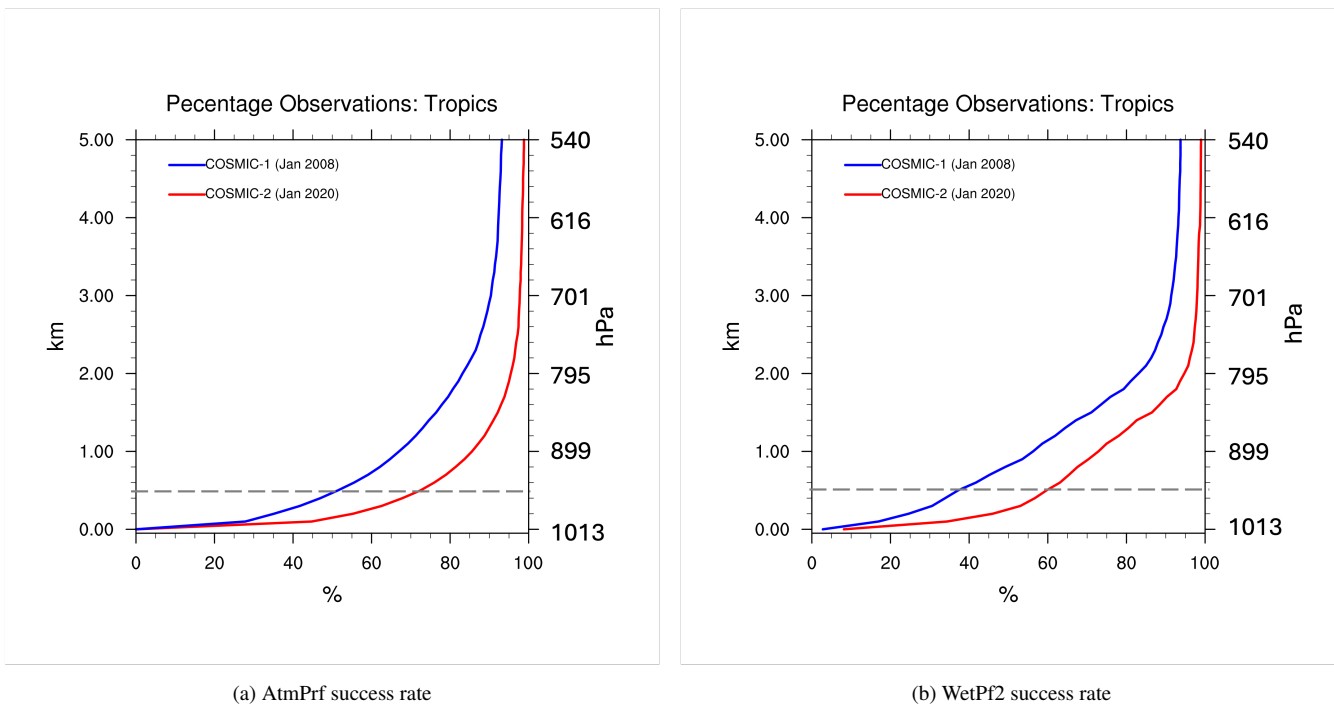

(a) AtmPrf success rate                            (b) WetPf2 success rate

**Figure 1.** Level 2 atmPrf (temperature) and wetPf2 (water vapor) successful retrieval rate (%) as a function of height above sea-level from COSMIC-1 during January 2008 (blue) and COSMIC-2 during January 2020 (red). Success rate is calculated by dividing number of valid GNSS-RO retrieval files over number of Level-1B file at a certain height. The gray dashed lines mark the reference at 0.5 km from the tropical ocean surface.

Wu et al. (2022) found out that the Level-1B deep SNR from the straight-line height ($H_{SL}$) is statistically significantly correlated with the MABL water vapor amount in the European Centre for Medium-Range Weather Forecasts (ECMWF) Reanalysis v5 (ERA-5) after averaging over a month at $2.5° X 2.5°$ grid resolution. The averaging is necessary to effectively beat down the random noise. This paper attributed such a positive correlation to the strong refraction from a horizontally stratiform and dynamically quiet MABL water vapor layer that acts to enhance the SNR amplitude at deep $H_{SL}$ through ducting and diffraction/interference (a summary recapitulation of the physical mechanism can be found in Section 2.3). Some caveats of this work limit its application to weather phenomena. First, it builds upon a single level regression statistics, the correlation coefficient of which was found the highest at $H_{SL} = -100km$ in the tropics, and $H_{SL} = -80km$ at high latitudes. Hence, any simple linear regression-based retrieval algorithm will suffer from arbitrary latitudinal discontinuities. As a matter of fact, SNR at different $H_{SL}$ levels are found correlated with MABL water vapor with different signs and magnitudes (e.g., Fig. 2), which should be used together to enhance the information content. Secondly, the robust relationship is only found for monthly averages in Wu et al. (2022), because the profile-by-profile noise is usually too high to yield a meaningful retrieval from SNR, and only through averaging large amount of profiles can the noise be lowered down to the level where signal stands out. These are all caveats of traditional statistical approach. Machine learning approach, however, is suitable at picking up weak signals through large amount of training data. As such, the scopes of this paper are to demonstrate the feasibility of using the ML method to extract MABL WV information from the GNSS SNR signals, and to demonstrate the scientific value of this new product over the existing operational water vapor retrievals.

Artificial Intelligence/Machine learning (AI/ML) applications in remote sensing field is trending in the past decade. It has been increasingly used in remote sensing fields in recent years. Traditional physics-based radiative transfer (RT) theories and modelings are used to link the remote sensing measurements (e.g., GNSS radio occultation signal) to the physical quantities (e.g., temperature and water vapor profiles). They are often highly non-linear, computationally expensive and involving many explicit or embedded assumptions/simplifications, which may or may not propagate properly into part of the retrieval errors eventually. Given the fact that satellite measurements usually contain large amount of data, and the association is highly non-linear between the measurement space and the physical space, the retrieval process becomes an ideal testbed for ML capabilities. Some pioneer works had attempted this approach to retrieve PBL atmosphere profiles and achieved notable success. For example, Ye et al. (2021) used the routine radiosonde measurement at an Atmospheric Radiation Measurement (ARM) site as the ground truth to train a ground-based infrared spectrometer to predict the PBL height. The capability is limited to only the stations where both observations are routinely available. Milestein and Blackwell (2016) employed a neural network (NN) framework on retrieving the temperature and water vapor profiles from the spaceborne Atmospheric Infrared Sounder (AIRS) observations (AIRS Version 7 product). The training "truth" was from the ECMWF analysis fields. It is worth mentioning that Milstein (2022), as a follow-up work, pointed out the ML-only retrieval framework tends to smooth out sharp gradient features in proximity to the PBL top. To mitigate this caveat, Milstein et al. (2023) employs the 3D deep neural network training on the AIRS granule image against ERA-5 reanalysis that helps PBL height recognition from passive imagers.

In this paper, we will explore the ML capability at retrieving the MABL WV information from the deep SNR signal at profile-by-profile basis (i.e., Level-2 standard). Section 2 introduces the training and validation datasets as well as the model

structure; Section 3 presents the retrieval results and independent validation; Section 4 expands the discussion to the usage of this product in studying MABL water vapor climatology and diurnal variabilities; Section 5 summarizes the major findings and shortcomings of the current work that may be improved in the future.

## 2 Data and Model

This section introduces the training, validation and independent validation datasets, as well as the ML model architecture and the underlying physical foundations for the ML technique to root upon.

### 2.1 Training and Validation Datasets

The definition of SNR follows Wu et al. (2022) which uses the normalized SNR ($S_{RO}$):

$$S_{RO} = (SNR - \sigma)/(SNR_0 - \sigma) \tag{1}$$

$$\sigma_S^2 = VAR(S_{RO} - \overline{S_{RO}}) \tag{2}$$

$SNR_0$ is the free atmosphere SNR. In practice, we use averaged SNR between 35 and 65 km altitude range as the $SNR_0$, and any profile with $SNR_0 < 200$ or $\sigma_{SNR_0}^2 > 0.05$ is considered "low-signal" and is filtered out. $\sigma$ is the instrument-specific noise determined for each individual instrument from very deep $H_{SL}$. The value for $\sigma$ used in this work is an updated version from Table A1 in Wu et al. (2022) and shown in the Appendix A (Table B1). Wu et al. (2022) also found an instrument-dependent shift of the mean $S_{RO}$ profile as a function $H_{SL}$. Luckily, such an issue can be resolved to use the excess phase at L1 ($\phi_{L1}$) as the vertical coordinate. In practice, the raw calculated $S_{RO}$ and $\sigma_S^2$ are mapped to a fixed 52-level $Log_{10}(\phi_{L1})$ vertical grid. It is roughly linearly correlated with $H_{SL}$. The value for the vertical grid is listed in Table B1 in the Appendix A. In practice, we also filtered out bad open-loop profiles, profiles with data gap greater than 2 km, and profiles with outlier $S_{RO}$ or $\sigma_S^2$ values.

The ERA-5 reanalysis is so far the best global reanalysis dataset in terms of PBL water vapor amount and distribution. Johnston et al. (2021) compared specific humidity from ERA-5 and MERRA-2 reanalysis against collocated and coincident GNSS-RO wetPf2 specific humidity retrieval profiles, and found ERA-5 outperforms MERRA-2 everywhere in the PBL. They both exhibit consistent dry biases with larger bias from mid-high latitudes. However, ERA-5 percentage bias is roughly 1/2 of that from the MERRA-2 reanalysis in the PBL and tropopause layers. Given that many previous works used ERA-5 reanalysis or ECMWF analysis for training or validating the satellite retrievals for water vapors (e.g., Milestein and Blackwell (2016), Milstein et al. (2023)), especially some recent ones using it as the standard to evaluate recent GNSS-RO missions (e.g., Chang et al. (2022), Zhran (2023), Ganeshan et al. (2025)) , it is well justified to use ERA-5 hourly reanalysis as the "training" dataset to create a large sample globally. However, it is also warned in Johnston et al. (2021) that GNSS-RO retrievals tend to have its own biases especially in MABL, and in fact some other research suggested wet biases in certain regions (e.g., Virman et al. (2021).)

In this work, we created a collocated and coincident ERA-5 - SNR training and validation dataset. The SNR records are from four satellite series: COSMIC-1, COSMIC-2, Metop-A and Metop-B. The periods for training, independent testing,

and prediction are listed in Table 1. Note that the testing period is independent from training period to avoid potential self-correlation using standard random splitting procedure. The prediction period however covers both training and validation
periods mainly for generating enough samples to construct statistically robust climatology (e.g., diurnal cycles). This however creates an unfortunate data leakage concern (e.g., as pointed out by Kapoor and Narayanan (2023)) for the comparison with the MAGIC campaign but not for the rest of other independent validation datasets (Table 2). The target variables are specific humidity at the aforementioned 6 pressure levels ($975hPa$, $950hPa$, $925hPa$, $900hPa$, $875hPa$ and $850hPa$). The input parameters are 52 levels of $S_{RO}$, 52 levels of $\sigma_S^2$, latitude, longitude, month and Rising/Setting flag.

**Table 1.** Training, testing and prediction periods

| Training (90% and 10% random-splitting) | COSMIC-1 | 2012.01-2012.12, 2016.01-2016.03, 2017.01-2017.03 |
|---|---|---|
| | Metop-A | 2017.01-2017.03 |
| | Metop-B | 2017.01-2017.03 |
| Testing | COSMIC-1 | 2018.01-2018.03 |
| | Metop-A | 2018.01-2018.03 |
| | Metop-B | 2018.01-2018.03 |
| Prediction | COSMIC-1 | 2012.01-2012.12, 2013.01-2013.12, 2016.01-2016.03, 2017.01-2017.03, 2018.01-2018.03 |
| | COSMIC-2 | 2020.01-2020.12 |
| | Metop-A | 2012.01-2012.12, 2013.01-2013.12 |
| | Metop-B | 2013.02-2013.12 |

Fig. 2 elucidates the linear correlation between COSMIC-1 $S_{RO}$ at each of the 52 levels and ERA-5 specific humidity at 975, 950, 925, 900, 875 and 850 hPa over global ocean. The largest positive correlations are found around Level #40 to Level #45, which roughly correspond to $H_{SL} = -100km$ to $-80km$ (Table B1). Based on the monthly averages, Wu et al. (2022) found the highest correlation at $H_{SL} = -100km$ in the tropics and at $H_{SL} = -80km$ for the polar regions, which is consistent with our profile-by-profile correlation as well. But Fig. 2 also shows positive or negative correlations at different $Log_{10}(\phi_{L1})$ levels,
which impede methods like multi-variable linear regression from working. $\sigma_{SNR}^2$ also exhibits non-linear patterns with slightly weaker correlations with MABL water vapor that are opposite in sign compared to that of $S_{RO}$. It is worth noting that these relationships are also instrument dependent, as can be clearly seen in the $S_{RO}$ cross-correlation for Metop-A and Metop-B in the Appendix Fig. A1 and A2. Considering the instrument-dependent correlation patterns, three ML models are developed separately for COSMIC, Metop-A and Metop-B satellites, although it probably redundant to build two separate ML models

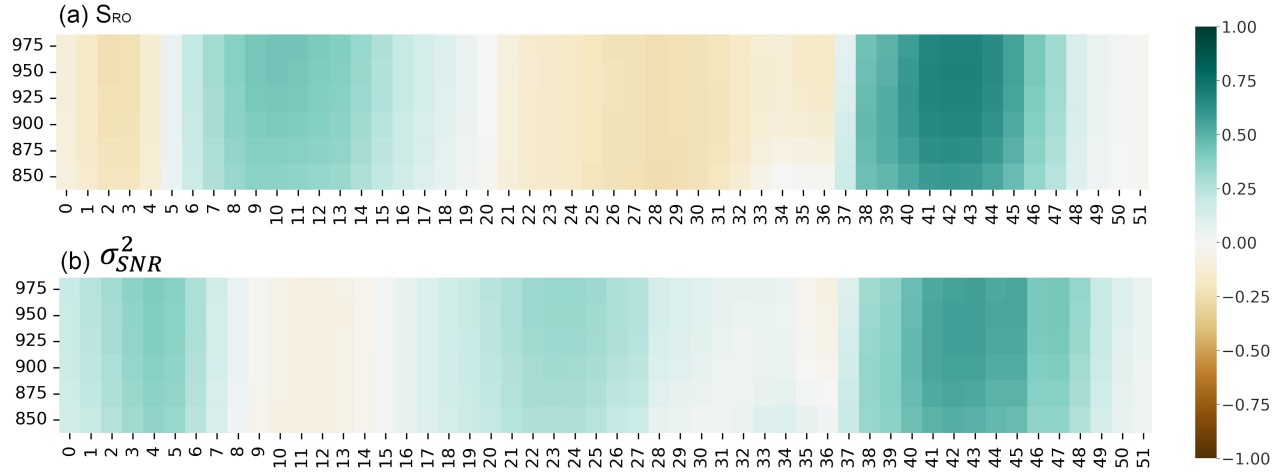

**Figure 2.** Correlation between collocated ERA-5 specific humidity at 975 - 850 hPa and $S_{RO}$ (top) and $\sigma^2_{SNR}$ (bottom) at various excess phase levels from the training COSMIC-1-ERA-5 dataset). Only grid indices are shown in the axis titles, and the corresponding $Log_{10}(\phi_{L1})$ values can be found in Table B1

.

for Metop-A and Metop-B separately as their correlation patterns are nearly identical. For the COSMIC series, we observed similar pattern from COSMIC-2 compared to Fig. 2 after downsampling the frequency to 1 Hz (not shown). Therefore, the ML model developed using COSMIC-1 observations is applied directly to the downsampled COSMIC-2 SNR observations. Through this practice we can also test the transfer learning among similar satellite series for the hope of stitching them together for longer record in the future research.

The correlation holds with the same slope at piece-wise level using individual profiles. For example, between SNR at $H_{SL} = -100km$ and ERA-5 specific humidity at $950hPa$, Wu et al. (2022) observed the near linear correlation with monthly averaged and gridded data, while we can see that the same slope is preserved at profile-by-profile level in Fig. 3. While this robust correlation proves that developing a Level-2 MABL specific humidity retrieval product using SNR profiles is feasible, the discernible larger noise at individual profile level versus month averages (Fig. 3d) suggests it is a challenging task. ML

method is hence introduced to tackle this highly complex regression problem.

    GNSS-RO operational water vapor retrieval product provided by the University Corporation for Atmospheric Research (UCAR) is employed to evaluate the quality of the SNR-ML retrievals. This operational product is called "wetPf2". Compared to an old 2013 processed "wetPrf" version, "wetPf2" has better penetration depth (Wee et al. (2022)) and is used for constructing Fig. 1, but "wetPrf" product is used for the MAGIC campaign comparison because of data availability constraints at the

time when this research was conducted. We've compared the success rate in the MABL between wetPrf and wetPf2 during Jan. 2008 (Fig. 1b) and only found very marginal improvements for COSMIC-1. Note that the key Level-2 profile to enable the 1D-VAR retrieval used by the wetPrf/wetPf2 product is the bending angle, which is assimilated in the ERA-5 reanalysis.

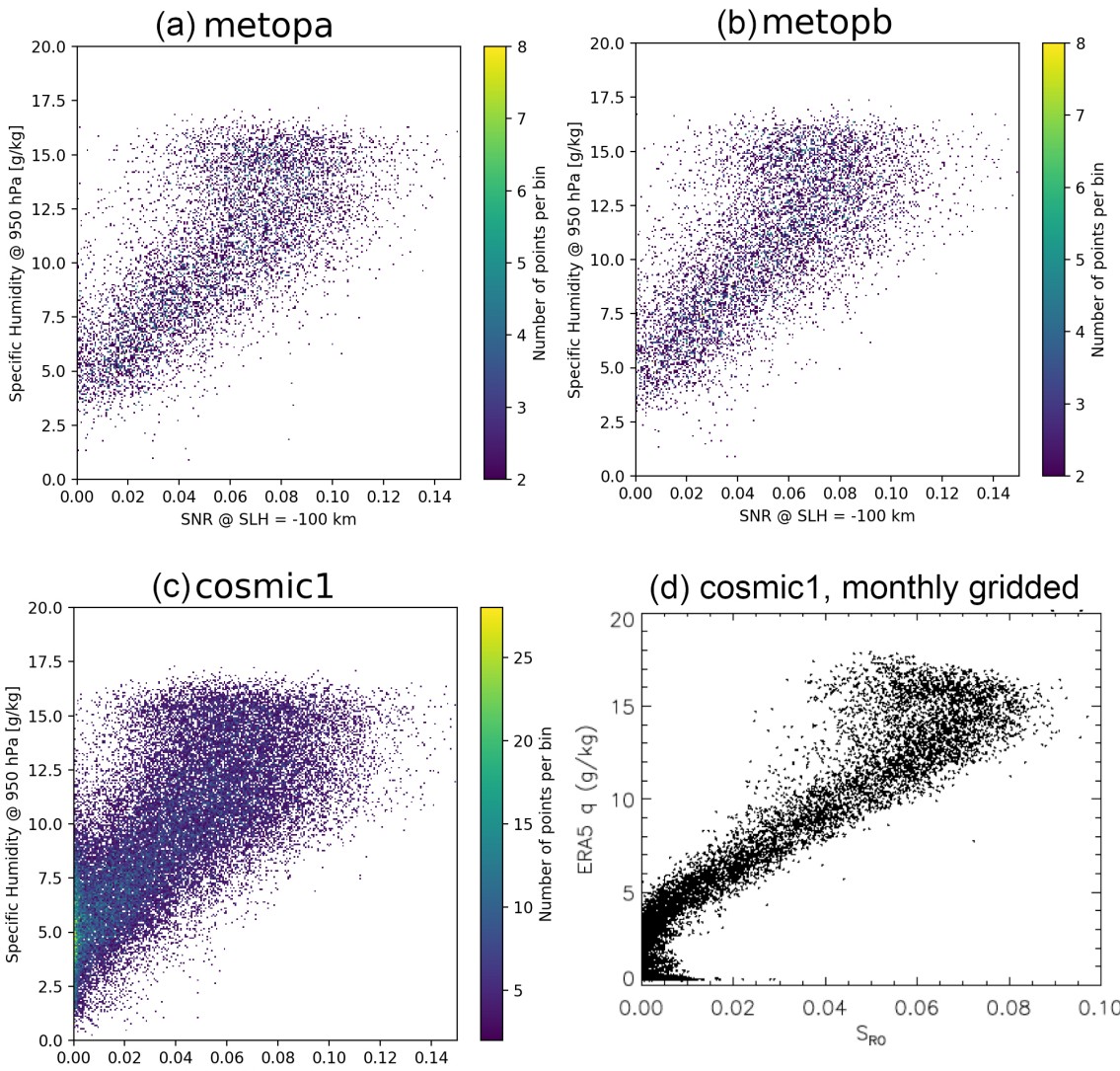

**Figure 3.** Density plots of the SNR-specific humidity relationship for (a) Metop-A, (b) Metop-B, (c) COSMIC-1 constructed from the entire training dataset between $45°S$ and $45°N$. The SNR value is taken from $H_{SL} = -100km$ while the specific humidity value is taken at $950hPa$. Fig. 9c from Wu et al. (2022) is reproduced here as (d) to demonstrate that the same relationship with the same slope holds at individual profile level.

Therefore, this is not an independent evaluation dataset. The purpose of this comparison is to identify the merits and caveats of the SNR-ML retrievals against an existing mature product.

In addition to the independent testing which is a standard procedure for ML/AI training and evaluation against the wet-Prf/wetPf2 operational product, a handful of shipborne radiosonde campaigns and airborne dropsonde campaigns data are

collected for further independent assessment. The campaign names, location and total number of valid profiles are presented in Fig. 4 and Table 2. We can see from the summary of weather scenarios during each campaign that this independent validation dataset comprehensively covers major marine weather regimes from extremely dry Southern Ocean (MARCUS), mid-latitude
stratocumulus region (MAGIC), tropical trade cumulus region (EUREC4A, ATOMIC), to episodically wet atmospheric river events (ARRecon). This exercise is critical for assessing the quality of ERA-5, wetPrf/wetPf2 retrieval, and Level 1 SNR-based retrieval under different weather scenarios. Moreover, as the ML model trained solely on COSMIC-1 SNR data is then applied to the COSMIC-2 data, the independent validation using the three campaigns in 2020 (ARRecon-2020, EUREC4A and ATOMIC) provides some solid evidences to evaluate the robustness of the "transfer learning".

**Table 2.** Campaign Information

| Campaign Name | Period used for validation | Location | Weather Regime | Type | Reference |
|---|---|---|---|---|---|
| MARCUS | 2017.11-2018.03 | Southern Ocean | Mixed-Phase PBL cloud | Radiosonde | Evan et al. (2022) |
| ATOMIC | 2020.01-2020.02 | Tropical North Atlantic | Tropical trade wind zone | Radiosonde and Dropsonde | George et al. (2021) |
| EUREC4A | 2020.01-2020.02 | Tropical North Atlantic | Tropical trade wind zone | Radiosonde | Stephan et al. (2021) |
| MAGIC | 2012.10-2013.09 | Eastern North Pacific Ocean | Subtropical MABL | Radiosonde | Evan et al. (2022) |
| ARRecon | 2018.02; 2020.01-2020.02 | Northeast Pacific off the coast of California | Atmospheric River | Dropsonde | Zheng et al. (2024) |

## 2.2   Machine Learning Model Selection

The Convolutional Neural Network (CNN) model (LeCun et al. (2015)) is chosen as our regression ML model. The model internal architecture is illustrated in Fig. 5. There are a total of 109 input parameters, including one dimensional array of $S_{R0}$ of 52 elements, one dimensional array of $\sigma_S^2$ of 52 elements, both interpolated to a fixed excess phase grid (Table B1), and latitude, longitude, month and Rising/Setting flag. The output parameters are specific humidity at 6 pressure level between $975$
and $850\ hPa$ with cadence of $25\ hPa$.

Compared to some earlier ML models (e.g., random forest, gradient boosting), CNN learns also the vertical cross-correlation within the 52-layer input SNR profiles, as well as within the targeted 6-layers of specific humidity profiles. We conducted a comprehensive search of best hyperparameters using the root-mean-square-error (RMSE) as the loss function.

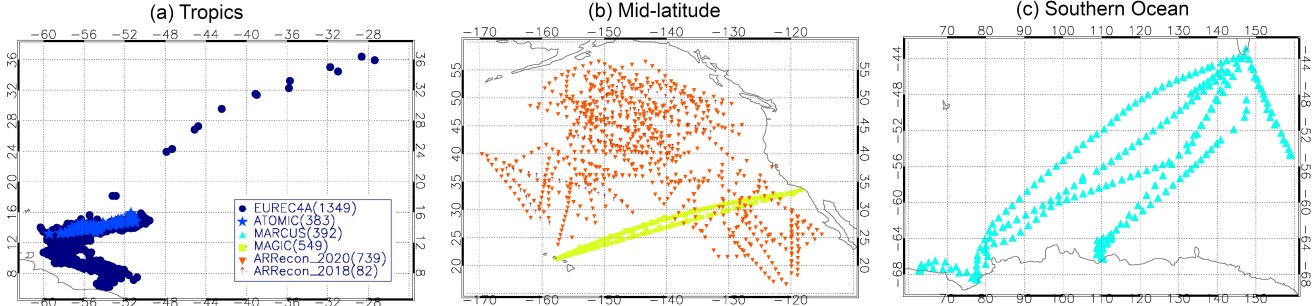

**Figure 4.** Maps for radiosonde/dropsonde locations from different shipborne or airborne campaigns in (a) tropics; (b) mid-latitudes; (c) southern ocean. Detailed campaign information can be found in Table 2. The total number of valid radiosonde/dropsonde profiles are listed in the parentheses in the legends.

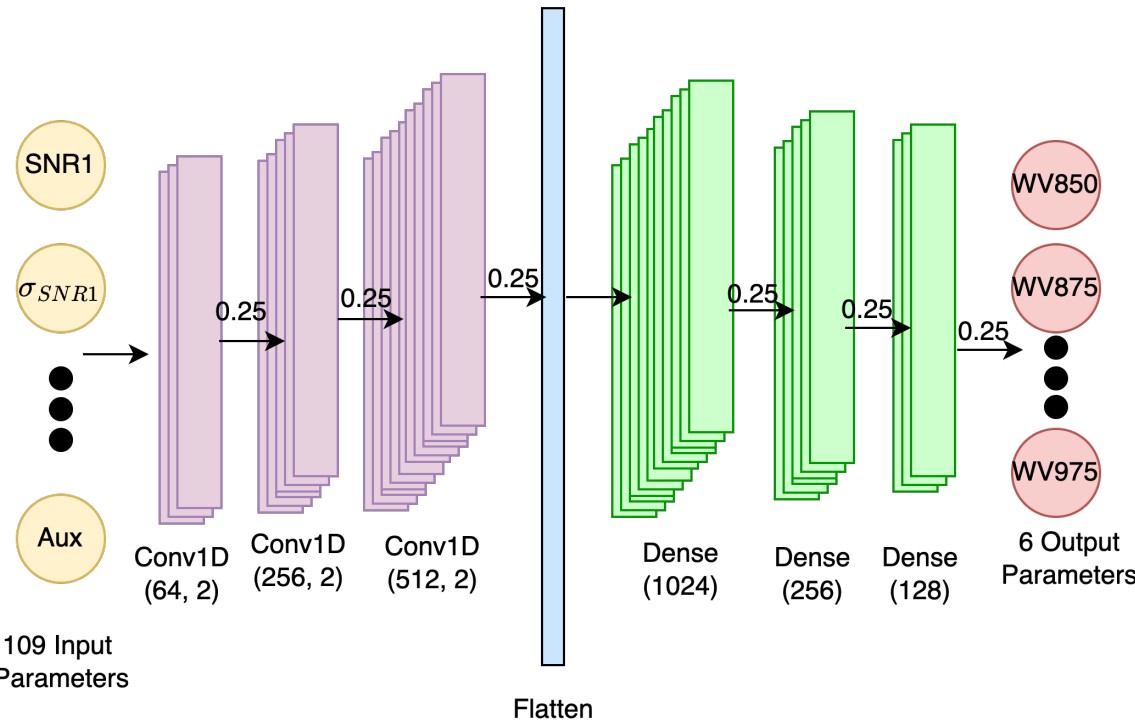

**Figure 5.** CNN model internal structure for this work. The numbers on top of the right-pointing arrow is the Monte Carlo dropout value applied between each layer. Numbers inside the parentheses of Conv1D layer indicate filter size and pool size, while numbers inside the parentheses of the Dense layer indicate number of the fully-connected nodes. The training takes 100 epochs, which suffice the needs for quick convergence.

In the prediction step, 30 predictions were carried out given each input set of variables, the mean and standard deviation of which were used as the final prediction and estimated uncertainty. It is worth highlighting that in each convolutional and fully-connected layer, a dropout rate of 0.25 is applied to generate the variation, which is then used to calculate the standard deviation of the "ensemble prediction" as a way to measure the retrieval uncertainty. This so-called "Monte Carlo" dropout method was designed in ML as a standard technique to regularize model over-fitting (Srivastava et al. (2013)), but were also employed widely as a Bayesian-approximation to quantify model uncertainties (Gal and Ghahramani (2016)). Admittedly the current method only provides a quantification for ML model errors. There is no consideration of SNR measurement errors nor propagation of the error to the final retrievals at this moment, although this is certainly some procedure to be in place in the future works.

We also tried some earlier ML models, e.g., random forest (RF), gradient boosting (GB), support vector machine (SVM) from the scikit-learn library and one deep learning model multilayer perceptron (MLP) from the pytorch library. The model performances are actually very close in terms of evaluating the RMSE except for the SVM, the latter of which performed discernibly worse than the rest ML models. It is not a surprise finding as this is a relatively simple and straightforward task that ML models should handle easily, but not the case for multi-variable linear regression type of logistic models (hence, it explains the poor performance of SVM). As the main focus of this paper is science and new information content embedded in SNR signals, we will not deviate the attention to spend more time discussing these model results. The semi-transparency of RF and GB models is appreciated by us though. We compared the feature importance rankings with Wu et al. (2022) findings, and find high consistencies (e.g., high ranking of SNR at $H_{SL} = -100\ km$ in the tropics, and SNR at $H_{SL} = -80\ km$ ranks the top in the polar region).

### 2.3 Underlying Physical Mechanisms

It is necessary to provide a summary of the underlying physics to emphasize the solid physical ground for this product, so readers would not misunderstand this as a pure statistics-based ad-hoc finding. The underlying physical mechanisms to explain the observed high-correlation between MABL water vapor and the GNSS SNR signal remains to be an active research area. Wu et al. (2022) articulated that the diffractive effect on the RO signal under the condition of limb sounding through a sharp MABL can extend the signal below the sharp edge of the obstacle with a limited depth.

Both diffractive and refractive processes are required to happen along the radio wave propagation to produce the RO signal at deep $H_{SL}$. Another example ( Sokolovskiy et al. (2024)) found enhancement of SNR when super-refraction happens. In reality, complex MABL can produce a mixed effect in the soundings from a combination of conditions that include normal bending, grazing reflection, super-refraction, ducting or diffraction (Sokolovskiy et al. (2014)). As a result, sophisticated physical radiation transfer models (e.g., radiohologram, canonical transform) can in principle be used but at the expense of high computational costs and hence impractical operationally. Moreover, the retrieval itself is essentially still an under-constraint problem, which commonly occurs for satellite retrievals and assumptions (no matter physically making sense or not) need to be made to fully constrain the physical model. As the quasi-linear relationship is preserved at profile-by-profile level with larger

noise compared to the monthly gridded and smoothed data (Fig. 3), and the height-dependency of the regression coefficient is highly non-linear (Fig. 2), a ML model is simply the best choice to extract the signal.

## 3  Results

### 3.1  Retrieval Performance Evaluation

As the first comparison, Fig. 6 and Fig. 7 showcase the statistical closeness to the $1:1$ line and the resemblance of geographical distributions for the three independent testing months: January - March, 2018, for COSMIC-1. All 6 pressure levels are compiled together to make Fig. 6, but were otherwise look extremely similar if plotting level-by-level. The only deviation from $1:1$ line occurs at very small specific humidity values (ERA-5 specific humidity $< 1g/kg$), i.e., very dry conditions, normally occurs at high-latitudes.

Such a discrepancy reveals itself more clearly when we map out the percentage difference (Fig. 7b). The largest percentage differences indeed are found at polar regions as well as near the coastal lines with SNR-ML retrieved humidity tending to be larger than ERA-5. Note that to satisfy ducting or other diffraction conditions in order to use SNR signal at deep $H_{SL}$, the surface is required to be flat. Therefore, the discrepancies around the coastal line are believed to be related to issues with SNR-ML retrievals when topography starts to play a role. However, as we will show later in Fig. 10, ERA-5 indeed shows consistent dry-bias at high-latitudes compared to independent radiosonde measurements. So SNR-ML retrieval might produce a closer-to-truth results as will be seen later as well. Moreover, one can visually discern discrepancies in the tropical deep convection/ITCZ zones where ERA-5 in general is wetter than SNR-retrieved values. Such a discrepancy is not conspicuous in Fig. 7b simply because of the large value in the denominator. We will also show later that none of the three datasets we will evaluate (SNR-ML retrieval, GNSS-RO wetPrf/wetPf2 retrieval and ERA-5 reanalysis) capture well the tropical MABL structures. For the SNR-ML method, it is probably because the ducting assumption is easily and frequently violated in the tropical MABL.

### 3.2  Uncertainty Quantification

Unfortunately, for the very dry conditions, SNR-ML method retrieved specific humidity also inherently comes along with large uncertainties, as can be clearly seen in Fig. 8. The SNR signal is too weak in this situation to yield any robust retrievals even with powerful ML models. Although we still believe the SNR-ML retrievals might be "more correct" than ERA-5 for very dry conditions, in practice we mark any retrieval with greater than 50% uncertainty with a quality flag in the published product, and those data do not pass the the quality control to be used later in this paper for independent validation nor constructing the climatologies. This threshold only filters out about 2% of the data with very weak SNR signals. If we would apply a threshold of 20%, about 16% of data would be filtered out. In the later section when the diurnal cycle is compared using multi-year regional averaged data, we found that heavy-averaging effectively beats down the noise so to reveal a visible diurnal signal in the extremely dry polar region, whereas ERA-5 is essentially a fixed value (Fig. 14b). We can also see from Fig. 8 that almost

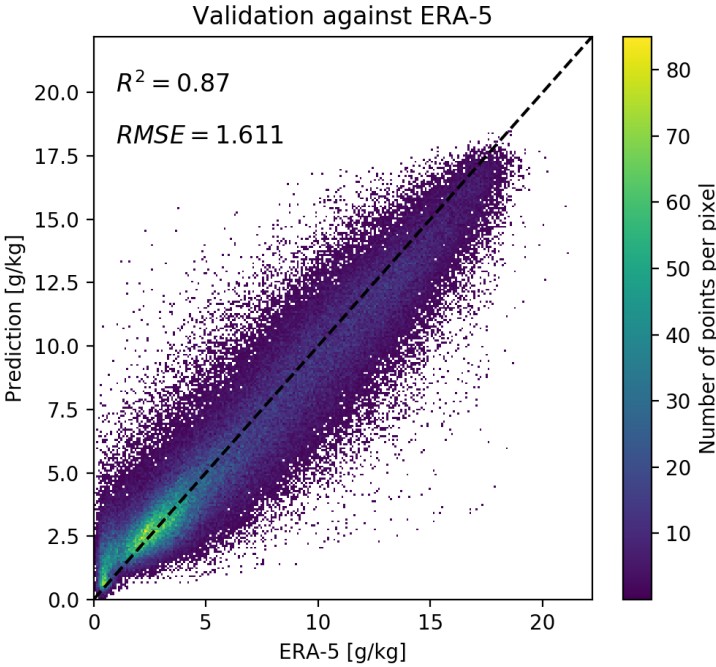

**Figure 6.** Heatmap for independent validation from January - March 2018 for COSMIC-1 combining all 6 levels together.

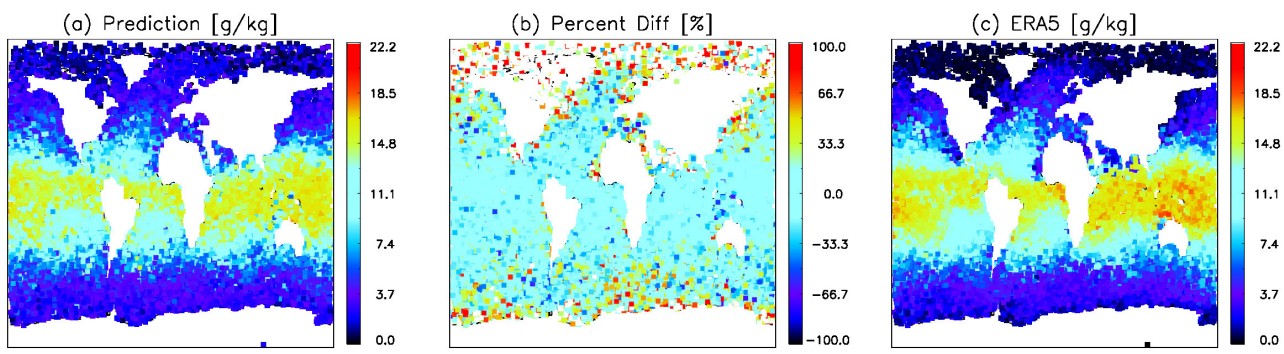

**Figure 7.** Geographic distribution of the (a) predicted values using COSMIC-1 SNR observations versus (c) ERA-5 validation values at 950 hPa for January - March, 2018. The middle panel is the percentage difference between (a) and (c).Only ERA-5 samples that collocate and coincident with COSMIC-1 SNR-ML retrievals are selected for this comparison

all SNR-ML retrievals greater than $2g/kg$ passes the quality control. Readers should keep in mind that our current uncertainty estimation approach under-estimates the real uncertainty because it does not take into account SNR errors.

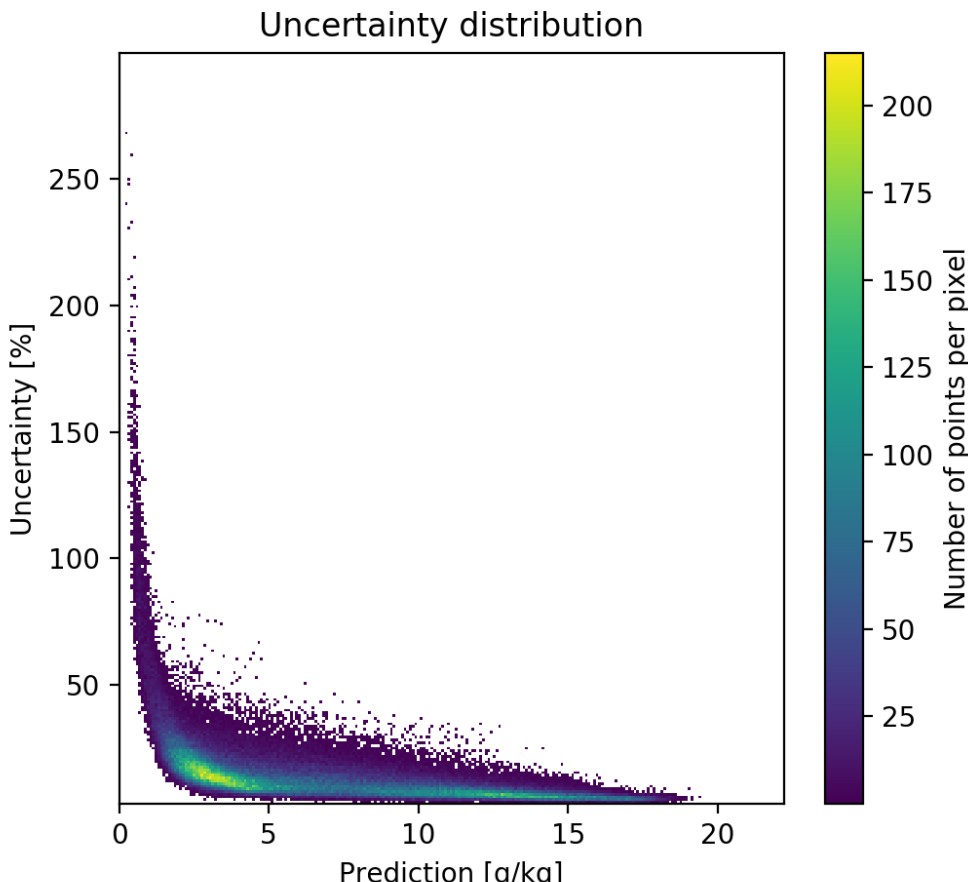

**Figure 8.** Percentage uncertainty distribution as a function of predicted value.

## 3.3 Comparison to Independent Radiosondes

In order to find collocation samples in every campaign, the collocation criteria are slightly different given the consideration of (1) the abundance of radiosonde/dropsonde profiles; (2) the typical spatial and temporal homogeneity of the local weather regime; (3) the availability of daily COSMIC-1, COSMIC-2, Metop-A and Metop-B profiles. In practice, for EUREC4A and ATOMIC, collocation is defined as longitude difference within $2°$, latitude difference within $1.5°$, and time difference within $1\ hr$. For the Southern Ocean campaign, the thresholds become $4°,2.5°$ and $2\ hr$ correspondingly. For ARRecon and MAGIC campaigns, the thresholds are $4°,1.5°$ and $2\ hr$.

Fig. 9 shows the level-by-level comparison for all collocated samples from all campaigns. SNR-ML retrieval results are shown in filled color symbols while wetPrf/wetPf2 retrievals are shown in open symbols. In addition, the averages from

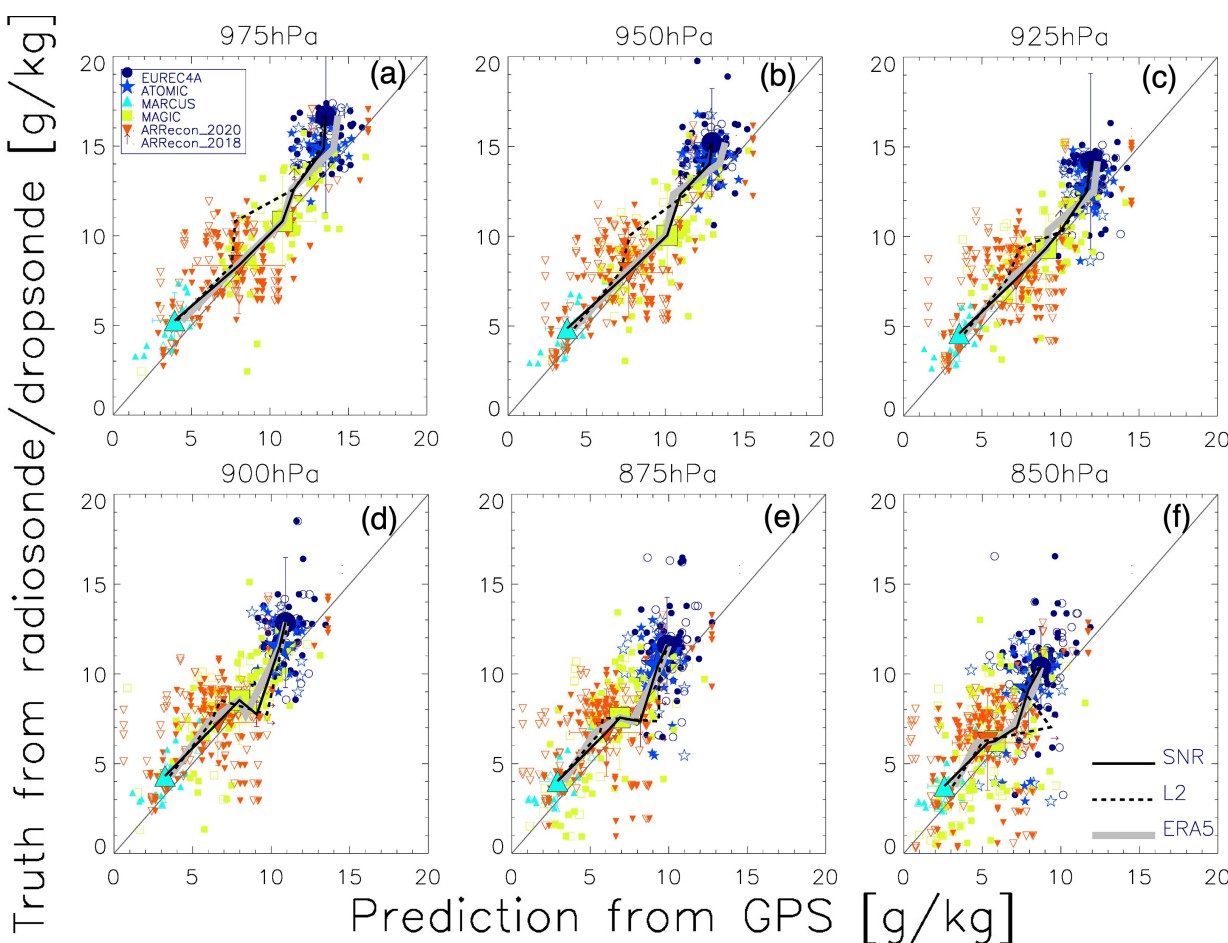

**Figure 9.** Scatter plots of collocated specific humidity [g/kg] comparison between radiosonde "truth" and retrievals from SNR (closed symbols) and wetPrf/wetPf2 standard retrieval (open symbols) for each pressure level. Black thin diagonal lines are the 1:1 lines for reference. The mean and standard deviation from the SNR-ML retrieval from each campaign are shown as bigger same color symbols with black boundaries. In addition, these mean retrieved values from each campaign are connected by the bold black lines for SNR-ML retrievals, bold black dash-dotted lines for wetPrf/wetPf2 retrievals, and bold gray solid lines for ERA-5 from the subset where collocations are found for SNR-ML and radiosonde data samples.

each campaign collocation subsets are connected together for better visual comparison against the $1:1$ lines (black solid lines for SNR-ML retrieval and black dotted lines for wetPrf/wetPf2 retrievals). We can see both SNR-ML retrievals and wetPrf/wetPf2 retrievals demonstrate generally good agreement with ground "truth" for different weather regimes. For the SNR-ML retrieval results, better correlations are found for the Southern Ocean (MARCUS campaign) and stratocumulus weather regimes (MAGIC campaign). Although wetPrf/wePf2 results are highly comparable to the SNR-ML retrievals, the collocation samples are much sparser for the former (Table 3). This could be attributed to the frequent occurrence of super

refractions in the stratocumulus region that causes a sampling bias of the wetPrf/wetPf2 results (Xie et al. (2010), Feng et al. (2020)). Spreads are slightly larger during the atmospheric river events (ARRecons). SNR-ML retrievals show an overall better agreement compared to the wetPrf/wetPf2 retrievals at all 6 pressure levels, especially for the few extremely large specific humidity values ($> 12g/kg$). The means of all ARRecon collocated samples also suggest that SNR-ML retrieval is the only one that does not produce a bias, while wetPrf/wetPf2 are moderately (slightly) dry biased in atmospheric river scenarios

at $> 900$ ($< 900$) $hPa$. ERA-5 from each campaign (only considering samples that SNR-ML retrieval collocation is found) exhibits good agreement to the ground truth too. For the two deep tropics campaigns ATOMIC and EUREC4A, we can clearly see that none of the three datasets capture the humidity conditions in the MABL very well. They are all dry-biased, and ERA-5 reanalysis is slightly less dry-biased than GNSS retrieved values at $975\ hPa$ and $950\ hPa$. SNR-ML method achieves overall comparable performance to ERA-5, which is expected because the model is trained on ERA-5.

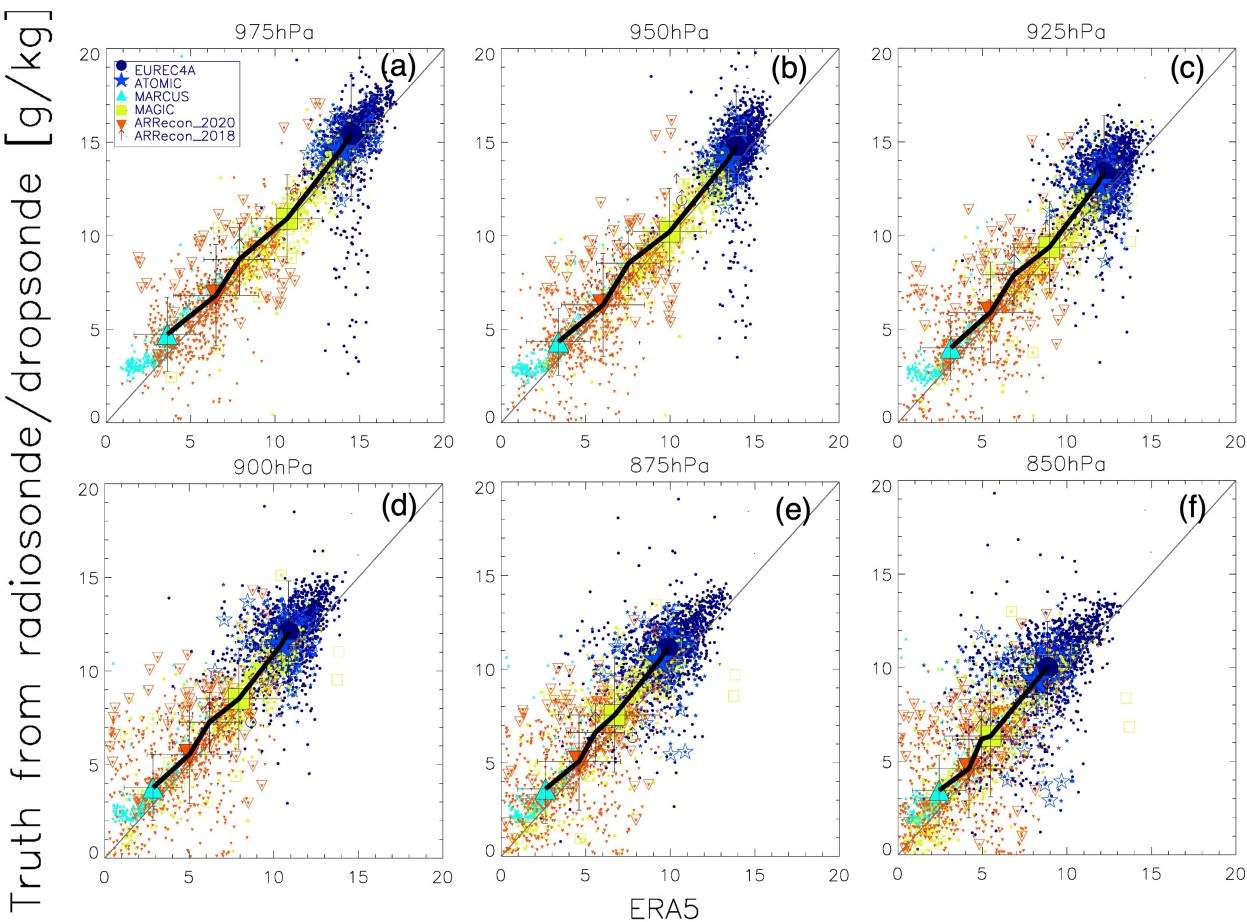

**Figure 10.** Same with Fig. 9, except for all available radiosonde/dropsonde samples in all these campaigns with collocated ERA-5 specific humidity. The means of each campaign are shown in bigger same color symbols with black boundaries and standard deviation. The bold black solid line connects the mean values from each campaign.

For convenience in pinpointing ERA-5 MABL issues, we also make Fig. 10 as each valid radiosonde/dropsonde profile from all 6 campaigns can always collocate with an ERA-5 reanalysis data sample within $1.5°$ longitude, $1°$ latitude and $1\ hr$ difference. Now we can clearly see ERA-5 frequently fails to produce the large variations in humidity in the trade-cumulus region (EUREC4A), the former of which tends to be always too wet. Otherwise, ERA-5 matches better than SNR-ML retrievals and wetPrf/wetPf2 retrievals in the deep tropics (EUREC4A and ATOMIC), albeit all of the three datasets contain persistent dry biases as also can be seen in Fig. 9. Another discernible bias happens in the Southern Ocean during the MARCUS campaign, where ERA-5 is consistently dry-biased when specific humidity is below $\sim 3g/kg$. The subset used to make the gray lines in Fig. 9 are overlaid in open symbols, so we can make straightforward and fare comparison between ERA-5 and SNR-ML retrievals. We can see that SNR-ML performs slightly better than ERA-5 in the atmospheric river scenarios (two ARRecon campaigns), and slightly worse than ERA-5 in the stratocumulus region (MAGIC campaign), both of which reflect in the correlation coefficient comparisons shown in Fig. 11 as well. Overall ERA-5 shows a small dry-bias globally at all levels, which agrees with early findings by Johnston et al. (2021) who used wetPf2 GNSS-RO retrievals to identify such a dry bias. Note that some of the campaign profiles (e.g., ARRecon dropsondes) are actually assimilated in the ERA-5 data, so it is not a completely independent validation strictly speaking. However, it is also worth noting that some previous publications employed ARRecon and EUREC4A radiosonde data as "ground truth" for evaluating ERA5 accuracy in capturing water vapor variabilities in the PBLs (e.g.,Cobb et al. (2021), Kruger et al. (2022)).

The violin plots in Fig. 11 and number of collocated sample statistics in Table 3 help disentangle the merits/caveats of SNR-ML retrievals from multi-dimensional statistical metrics. Only correlation coefficients of all collocated samples collected from each campaign are displayed in Fig. 11. The ARRecon-2018 and ARRecon-2020 samples are further combined. From Fig. 11a, we can see again that the MABL specific humidity is not well captured in the tropics by either of the three datasets (EUREC4A and ATOMIC), but SNR-ML retrievals perform slightly better than the operational wetPf2 products in the deep tropics and trade-cumulus regions. In the rest three campaigns in the mid- and high-latitudes, they all agree very well with the radiosonde/dropsonde ground truths. ERA-5 reanalysis does the best job at high-latitude southern ocean (MARCUS) as well as the stratocumulus region (MAGIC), while in the atmospheric river regime, SNR-ML retrievals outperform the wetPrf/wetPf2 retrievals as well as the ERA-5 reanalysis. It is worth noting that SNR-ML retrievals perform slightly better than wetPrf/wetPf2 retrievals in the stratocumulus region (MAGIC) in both the medians and the top-heavy skewness of its distributions, which can be partially attributed to the scarcity of wetPrf/wetPf2 collocation samples in this weather regime and known bias in the Level 2 retrieved refractivity gradient (Xie et al. (2010)). For the polar region (MARCUS), although SNR-ML retrievals exhibit the lowest correlations among the three datasets albeit all correlations are statistically significant, it is inconclusive at this point to say that SNR-ML method is not suitable for the polar region. As a matter of fact, SNR-ML method generates the largest variabilities among the three when the PBL is extremely dry (Fig. 6), but the SNR in this situation is generally too weak to generate a robust retrieval (i.e., uncertainty too large compared to retrieved value). The retrievals from the SNR-ML method at dry polar winters contain more potentials (e.g., Fig. 14 as an example) if future GNSS missions could improve the SNR.

Fig. 11b demonstrates the robustness of the SNR-ML retrievals across all 6 PBL pressure levels. Although the highest positive correlations are always identified in ERA-5 and/or wetPrf/wetPf2 products, the medians of SNR-ML retrievals are

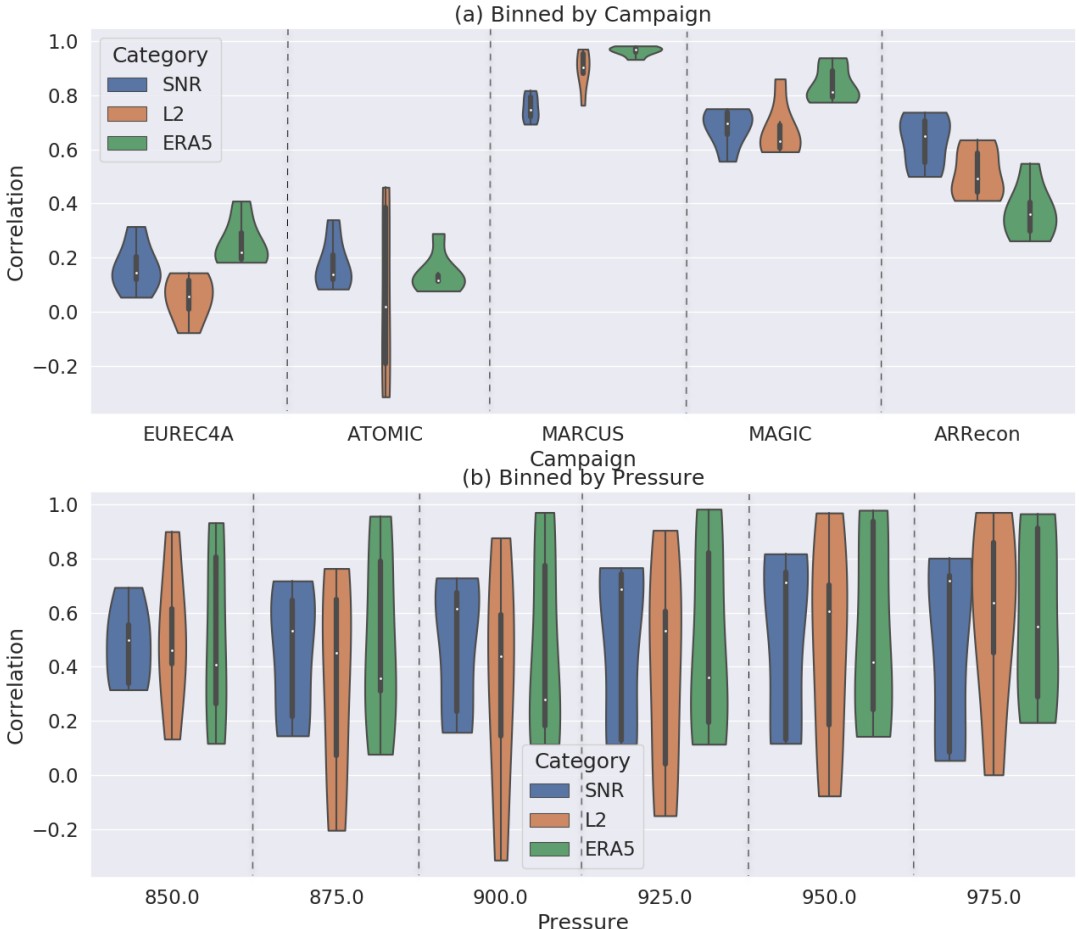

**Figure 11.** Violin plots of the correlation coefficients calculated from collocated samples for SNR-ML retrievals (blue), Level 2 retrievals (orange) and ERA-5 (green). (a) is all-level statistics for each campaign; and (b) is all campaign but binned by different pressure levels. Medium, standard deviation, minimum/maximum values and the skewness of the distribution are shown as the white dots, black box, extended vertical thin lines and the horizontal widths in each violin, respectively. The number of total samples are listed on top of each violin. For ERA-5, only the subset of samples that SNR-ML retrieval collocations available are selected to calculate the statistics.

consistently the highest with consistent top-heavy distribution except for 850 hPa, meaning that SNR-ML retrievals agree with radiosonde/dropsonde "truths" more consistently while ERA-5 and wetPrf/wetPf2 have more variations across different weather regimes. Of course all these conclusions are limited by the small collocation samples (<=309 in total), and we for sure need more extensive evaluation for this research product before massive production.

Another big advantage of the SNL-ML retrieval is its consistently higher success rate in the MABL compared with the wetPrf/wetPf2 product. This is clearly seen in Table 3, where the percentage difference between the two are listed in the parentheses for each campaign at each pressure level. For stratocumulus region (MAGIC campaign) that ducting or super-refraction happens frequently, the success rate of SNR-ML method can go up to 700% more than using the wetPrf product at the lowermost altitude. Although the superiority of success rate of the SNR-ML retrievals gradually vanishes when getting closer to the MABL top, they are still across-the-board more than wetPrf/wetPf2 products.

Table 3. Number of collocated GNSS-radiosonde/dropsonde samples in each campaign. Two numbers in each cell are from SNL-ML method and wetPrf/wetPf2 product, respectively, and their percentage differences are shown in the parentheses.

| Campaign Name | $975\ hPa$ | $950\ hPa$ | $925\ hPa$ | $900\ hPa$ | $875\ hPa$ | $850\ hPa$ |
|---|---|---|---|---|---|---|
| EUREC4A | 50, 19 (160%) | 50, 23 (117%) | 51, 29 (76%) | 51, 31 (65%) | 51, 34 (50%) | 51, 38 (34%) |
| ATOMIC | 49, 23 (113%) | 49, 27 (81%) | 49, 29 (69%) | 49, 29 (69%) | 49, 35 (40%) | 49, 44 (11%) |
| MARCUS | 13, 5 (160%) | 13, 7 (86%) | 13, 7 (86%) | 13, 7 (86%) | 13, 8 (63%) | 13, 9 (44%) |
| MAGIC | 72, 9 (700%) | 72, 25 (188%) | 72, 34 (112%) | 72, 40 (80%) | 72, 43 (67%) | 72, 46 (57%) |
| ARRecon | 120, 84 (43%) | 120, 101 (19%) | 120, 101 (19%) | 120, 106 (13%) | 120, 106 (13%) | 120, 106 (13%) |

To summarize the major findings for comparisons against the limited independent radiosonde/dropsonde datasets available over the open ocean, we can draw the following conclusions. Firstly, the quality of the SNR-ML retrievals is comparable to ERA-5 and the operational wetPrf/wetPf2 product. In atmospheric river weather regime, SNR-ML method even outperforms the other two. The robustness and stable performance of SNR-ML retrievals remain the best within the MABL, although its advantage gradually vanishes with increasing height. Secondly, compared to the operational retrievals, the SNR-ML method can achieve $10 - 700\%$ more samples in the MABL, especially over stratocumulus regions where ducting and super-refraction frequently occur that cause failure of operational retrievals. This suggests some unique value that the SNR-ML method can bring to the science community in facilitating understanding the water vapor-stratocumulus coupling mechanisms. Although some of the "independent validation dataset" is not completely independent as they may have been assimilated in the ERA-5, the fact that SNR-ML retrieval statistics outperform ER-5 at all 6 pressure levels in diverse weather regimes prove that real physical information from SNR observations is learnt and kept by the ML model for prediction, admittedly it is impossible to quantify how much the real observed information contributes without accurate physics-based model simulations.

## 4 Discussions

In this section, we present and discuss some use case examples in order to demonstrate how to use this SNR-ML MBPL specific humidity product to identify and even quantify model or reanalysis issues.

## 4.1 Climatology

Several previous studies suggest that MERRA-2 reanalysis has larger dry-biases in the polar regions compared to ERA-5 (Johnston et al. (2021), Ganeshan and Yang (2019)), while some other studies using in-situ campaign data suggested smaller dry-bias in the MERRA-2 reanalysis (e.g., Seethala et al. (2021)). Here we map out the climatological distribution of specific humidity retrieved using the SNR-ML method to track down geographical discrepancies in the Arctic (Fig. 12) and Antarctic (Fig. 13) with respect to MERRA-2. The coldest months were not selected because of the concern that sea ice induced reflectometry signal might contaminate our SNR-ML retrieval results, but we didn't exclude retrievals over possible glaciers that MERRA-2 do not produce a valid value at $925 \ hPa$ because we used a fixed terrain map. Therefore, direct comparison should not be considered wherever MERRA-2 value is blank.

Overall again we can see the SNR-ML method retrieved polar MABL is much more humid than that from MERRA-2 in the Arctic during early spring and late fall seasons ($> 100\%$ in most areas). If we neglect sampling induced geographical inhomogeneities in the SNR-ML retrievals, we can actually see in Fig. 12 that the geographic distribution of highs and lows and their gradients are in general agreeable. The largest difference is that the wet intrusion along the Bering strait seems to be too weak during both April and November in MERRA-2, which could account for the dry-bias in the deep Arctic ocean. Meanwhile, the wet intrusion associated with the North Atlantic overturning circulation seems to be too strong during November in MERRA-2. These discrepancies connect possible root causes down to the ocean circulation, and up to the Arctic front, and should be further investigated in a whole Earth-system point of view.

Although Southern Ocean and South Pole seem lacking geographical variations (Fig. 13), we can actually observe some interesting potential issues related to topographies. For example, the tip of the Andes mountain effectively blocks MABL water transport across the mountains, but such a local effect on humidity appears further downstream in MERRA-2. The gradient of water vapor amount from north to south is apparently much weaker compared to MERRA-2, which impacts the latent heat and sensible heat flux quantification when considering global energy transport.

## 4.2 Diurnal Variation

It is well-known that global climate models (GCMs) have serious issues at reproducing the cloud, precipitation and convection diurnal cycles (e.g., Tian et al. (2004), Yin and Porporato (2017)). Although such a problem is mostly attributed to the issues with cumulus parameterization schemes, we argue that the diurnal cycle of MABL water vapor also plays a nontrivial role as it ties closely to the shallow cumulus and stratocumulus, the latter, for example, is also closely related to the MABL height diurnal variation (e.g.,Liu and Liang (2010), Chepfer et al. (2019), Teixeira et al. (2021)). Ground truth of the diurnal variation of MABL water vapor structures is extremely rare, probably because of the high cost associated with long-duration shipborne campaign that often only launches radiosondes twice daily and hence cannot capture the diurnal variabilities. Therefore, here we only aim at showing the discrepancies between ERA-5 and our SNR-ML retrieval generated diurnal cycle rather than determining which is right or which is wrong (Fig. 14).

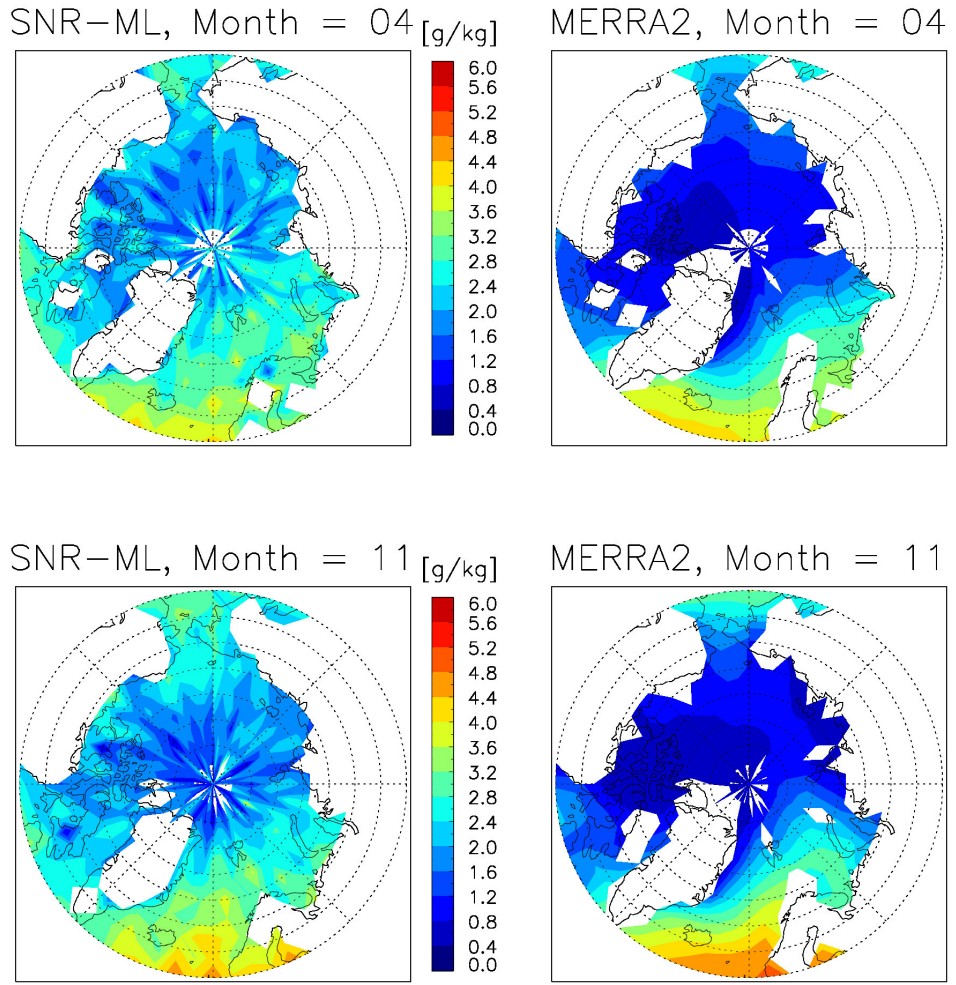

**Figure 12.** Monthly averages of 950 hPa specific humidity from COSMIC-1 SNR retrieval (left) compared to MERRA-2 reanalysis (right) for Arctic during April (top) and November (bottom), 2012 and 2013.

In addition to the Southern Ocean MARCUS campaign and the atmospheric river regime ARRecon campaign that we have ground truths to compare with, several additional campaign regions and corresponding months are selected motivated by the observed diurnal variations of the MABL height established in Liu and Liang (2010). These two additional regions include South Indian ocean (INDOEX campaign, representing deep tropics), and the Arctic open ocean, representing polar winter conditions. The last one was added for the sole interest to check if there is any diurnal cycle in the coldest season.

The averaged specific humidity at $875\ hPa$ agrees well between the two datasets in the MARCUS and ARRecon campaigns, but the diurnal cycles in ERA-5 are too weak compared to the ground truths (red asterisks), while the SNR-ML method retrieved

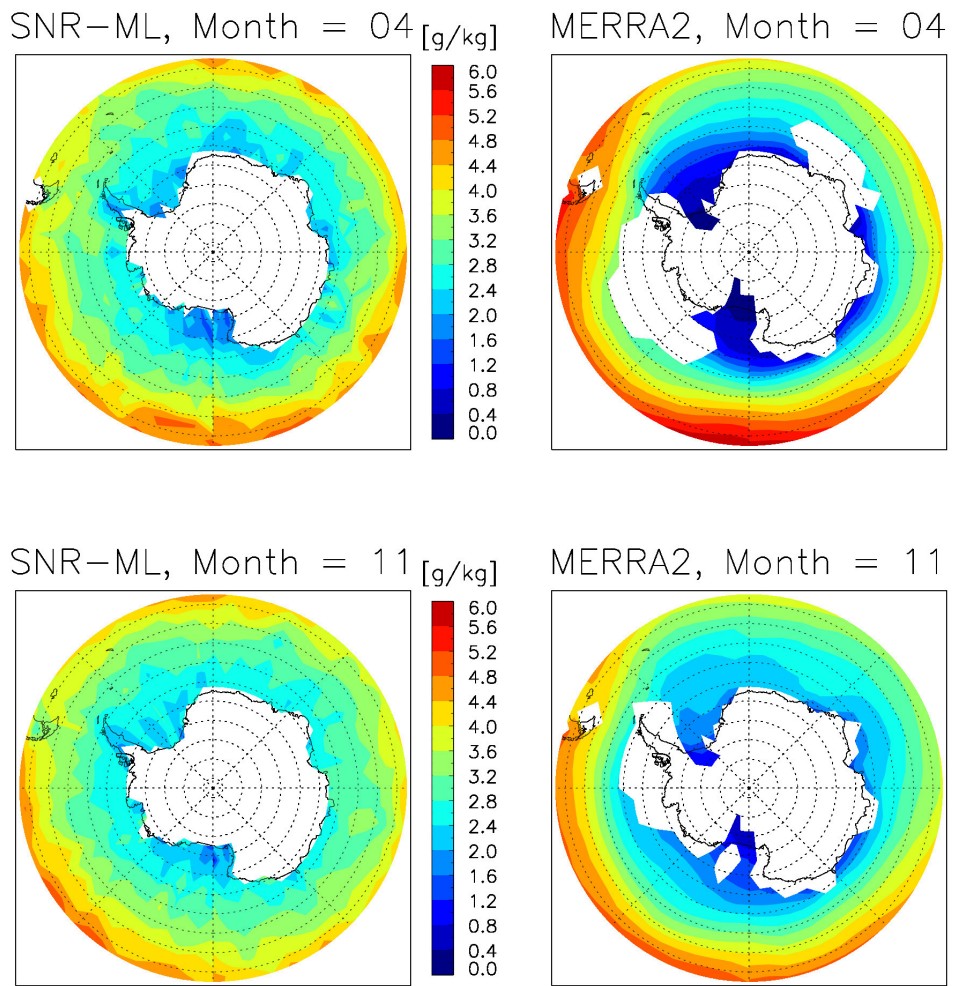

**Figure 13.** Same with Fig. 12, except for the Antarctic/Southern Ocean.

stronger diurnal cycles. It is worth noting that neither SNR-ML nor ERA-5 reproduced a strong peak below 900 hPa around
10 AM local time that both MARCUS and SOCRATES campaigns observed. The latter is another research campaign in the
vicinity of MARCUS ship routes and season (Vomel and Brown (2018)), but was not employed for independent validation
because of lack of collocations with GNSS observations. This peak is probably associated with the shallow mixed-phase cloud
pocket precipitation that is spatially so small and inhomogenous in scale (Alessandro et al. (2021)) that neither GNSS nor
ERA-5 are able to capture or reproduce. The under-estimation of the diurnal variability in the ARRecon campaign region
is probably associated with the sampling bias, because the campaign "truth" was sampled only during AR events, while the
SNR-ML and ERA-5 samples the climatology background.

Although we have no ground truth to assess the diurnal cycles of MABL humidity in other two regions, we can tell that ERA-5 is wetter in the South Indian ocean, and significantly drier in the Arctic ocean. Compared to the SNR-ML method retrieved diurnal cycle, MABL water vapor diurnal cycle in ERA-5 is too weak in 3 areas but not the INDOEX campaign region. To put into context of the diurnal cycle of PBL height (Liu and Liang (2010)), in the INDOEX campaign region, the diurnal cycle from ERA-5 and SNR-ML method agrees reasonably well, both anti-correlated with the diurnal cycle of PBL height change observed during that campaign. In the Arctic ocean, ERA-5 apparently has set some arbitrary threshold to keep the water vapor at a constant low level, while SNR-ML retrievals suggest a weak diurnal variation.

Overall, we can see the diurnal coupling between MABL water vapor, PBL height and clouds are vastly different from area to area. However, ERA-5 likely under-produces the diurnal cycle amplitude of MABL water vapor globally. For SNR-ML retrievals, day-to-day variability often overwhelms the signal of diurnal cycle, yet the amplitude of diurnal cycle is still stronger and matches better with the limited ground truth. Ultimately, the lack of MABL water vapor ground "truth" measurements will continuously make observing and verifying the true diurnal cycle difficult. Other shipborne measurements, e.g., upward pointing radiometers, might be helpful to disentangle this mystery in the future.

## 5 Conclusions

Marine planetary boundary layer (MABL) water vapor amount and vertical gradient are among the key factors to couple the ocean and atmosphere cloud, precipitation and convection together, but meanwhile it is also among the hardest objects to retrieve from satellite remote sensing perspective. Given the penetration capability of GNSS signal through clouds, we proposed a novel way in Wu et al. (2022) to utilize the GNSS signal-to-noise ratio (SNR) in the deep $H_{SL}$ to retrieve MABL water vapor profiles. In this paper, we demonstrated it is workable at profile-by-profile level, leveraging the power of machine learning (ML) in capturing weak and non-linear signals. The surprising and novel findings in this paper, is that the ML-trained model can make better predictions that outperforms the training dataset (i.e., ERA-5) in some places, which demonstrates that the real information content in the SNR signal is learnt which would otherwise not be harnessed using traditional statistical methods. The new SNR-ML retrieval has more stable performance against the operational wetPrf/wetPf2 GNSS-RO retrievals, and it can produce $20 - 700\%$ more successful retrievals in the lowest $1\ km$ where observations are critical to understand ocean-atmosphere exchange.

We then showed two use cases to demonstrate possible ways to use this dataset. There is no conclusive results because of lack of ground "truth" to validate, but we do find both reanalyses tend to systematically produce dry biases at high-latitudes, and too weak diurnal cycles over global oceans. This SNR-ML retrieval dataset also has its own caveats. Whenever the "ducting" condition is violated (e.g., coastal topography, convective tower, mixing and turbulence in the MABL), the fundamental assumption breaks down, resulting in poor performances. More extensive comparisons and validations against other high-quality ground measurements are needed in the future.

Based on results from this work, one can see that deep SNR can complement the current GNSS-RO operational bending angle product for retrieving PBL information for different PBL conditions. A merged product is certainly of interest to future

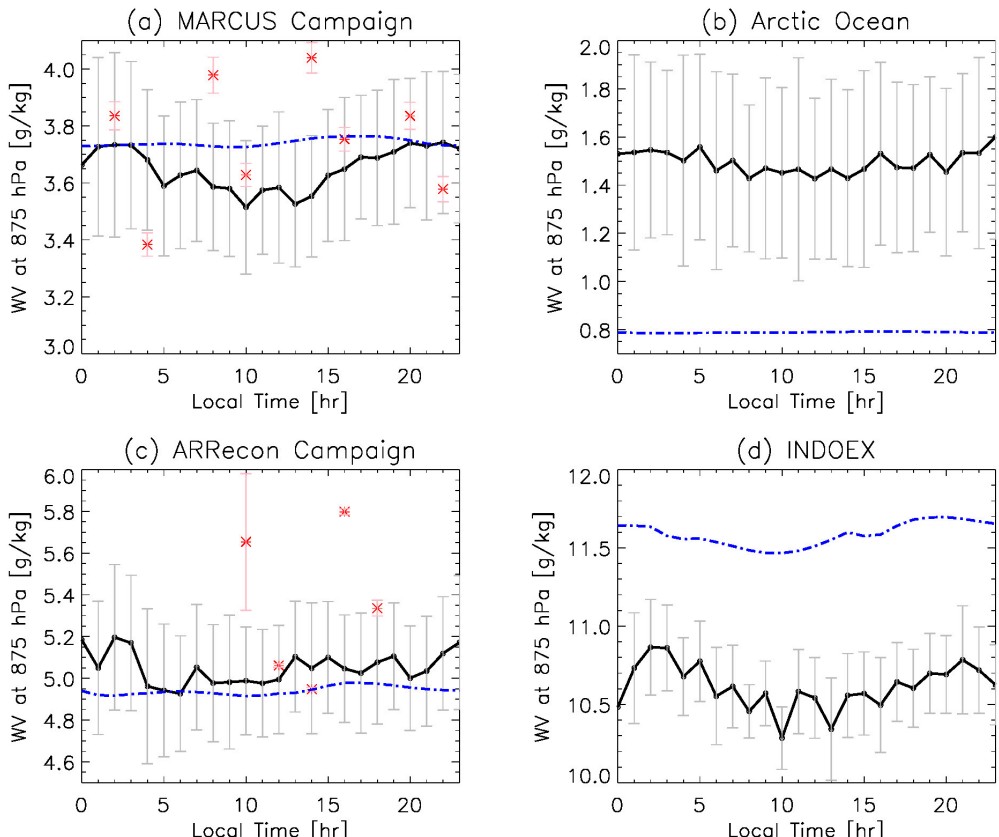

**Figure 14.** Multi-year mean diurnal variation of 875 hPa specific humidity retrieved from all four missions (black with errorbars in grey) and from ERA-5 hourly reanalysis (dash-dotted blue) during November - March for (a) MARCUS campaign region, $60°E - 150°E$, $60°S - 40°S$; (b) Arctic ocean, $180°W - 180°E$, $70°N - 90°N$; (c) ARRecon campaign region, $160°W - 120°W$, $20°N - 50°N$; (d) INDOEX campaign region, $55°E - 75°E$, $25°S - 15°S$. The MARCUS radiosonde and ARRecon dropsonde "truths" are overlaid in (a) and (c) as asterisks with standard deviations shown in pink vertical bars.

investigations, but fully understanding the physical mechanisms behind the reemerged deep SNR signal is the foundation for other downstream applications (e.g., data assimilation). Right now this can be considered as a stand-alone observational product for independent comparison or validation against model simulations or other observations.

*Data availability.* The Level 2 SNR-ML retrieval product for the prediction period (see Table 1) has been published on zenodo (Gong et al. (2024)). We welcome use and feedbacks.

COSMIC-1 and COSMIC-2 Level 1 and Level 2 data are downloaded from https://data.cosmic.ucar.edu/gnss-ro/. Metop-A and Metop-B data are downloaded from https://gpsmet.umd.edu/gnssro/download.php. ATOMIC data are downloaded from https://psl.noaa.gov/atomic/

data/. EUREC4A data are downloaded from https://doi.org/10.25326/137. SOCRATES data are downloaded from https://data.eol.ucar.edu/ master_lists/generated/socrates/. MARCUS data are downloaded from ARM data request portal. MAGIC data are downloaded from ARM data request portal. ARRecon data are downloaded from https://ARRecon.ucsd.edu/arrecon_data/ specially processed to fit the needs of this
research. Interested users are encouraged to contact the last author for assistance of post-processed data.

## Appendix A:  A

**Table A1.** Summary of GNSS-RO instrument noise ($\sigma$) used in this work, separated by rising and setting modes.

| Instrument Name | Orbit | Noise ($\sigma$) |
|---|---|---|
| COSMIC-1/C1 | Rising | 10.1 |
|  | Setting | 10.9 |
| COSMIC-1/C2 | Rising | 10.2 |
|  | Setting | 10.9 |
| COSMIC-1/C3 | Rising | 9.6 |
|  | Setting | 10.4 |
| COSMIC-1/C4 | Rising | 10.6 |
|  | Setting | 11.2 |
| COSMIC-1/C5 | Rising | 10.1 |
|  | Setting | 11.1 |
| COSMIC-1/C6 | Rising | 9.2 |
|  | Setting | 10.7 |
| COSMIC-2/E1 | Rising | 17.0 |
|  | Setting | 17.5 |
| COSMIC-2/E2 | Rising | 17.5 |
|  | Setting | 17.8 |
| COSMIC-2/E3 | Rising | 17.2 |
|  | Setting | 17.9 |
| COSMIC-2/E4 | Rising | 17.5 |
|  | Setting | 17.7 |
| COSMIC-2/E5 | Rising | 17.4 |
|  | Setting | 17.8 |
| COSMIC-2/E6 | Rising | 17.5 |
|  | Setting | 17.8 |

**Table B1.** Excess Phase L1 grid for this work

| Parmeter | Grid values |
|---|---|
| $Log_{10}(\phi_{L1})$ | 1.26245, 1.33846, 1.41162, 1.48144, 1.54777, 1.62428, 1.69679, 1.76530, 1.82995, 1.89098, 1.94866, 1.97000, 2.00325, 2.02000, 2.05500, 2.08000, 2.10415, 2.13000, 2.15091, 2.17000, 2.19548, 2.23805, 2.27875, 2.30103, 2.32222, 2.37000, 2.41497, 2.44000, 2.55630, 2.59000, 2.63000, 2.69020, 2.75000, 2.81291, 2.86000, 2.92428, 2.95000, 3.02531, 3.10000, 3.11727, 3.15000, 3.20140, 3.22000, 3.25000, 3.27875, 3.30000, 3.32000, 3.35025, 3.41664, 3.47857, 3.53656, 3.59106 |
| Rough corresponding $H_{SL}$ [km] | 19, 17, 15, 13, 11, 10, 9, 8, 7, 6, 5, 4, 3, 2, 1, -1, -2, -3, -5, -7, -9, -11, -13, -15, -17, -19, -20, -23, -26, -30, -33, -37, -40, -50, -60, -70, -80, -90, -92, -94, -96, -98, -100, -102, -104, -106, -108, -110, -120, -130, -140, -150 |

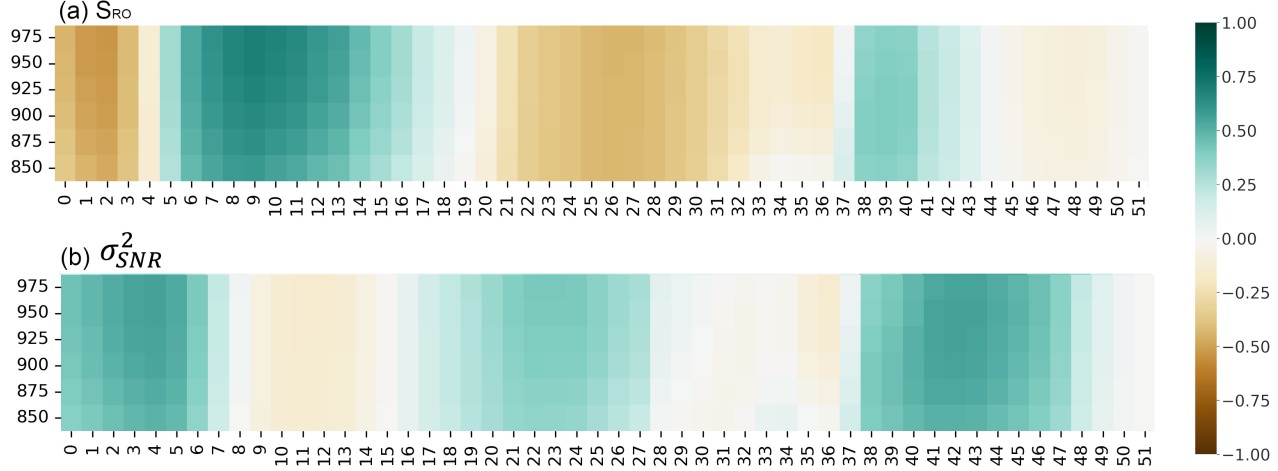

**Figure A1.** Same with Fig. 2, except for Metop-A training dataset.

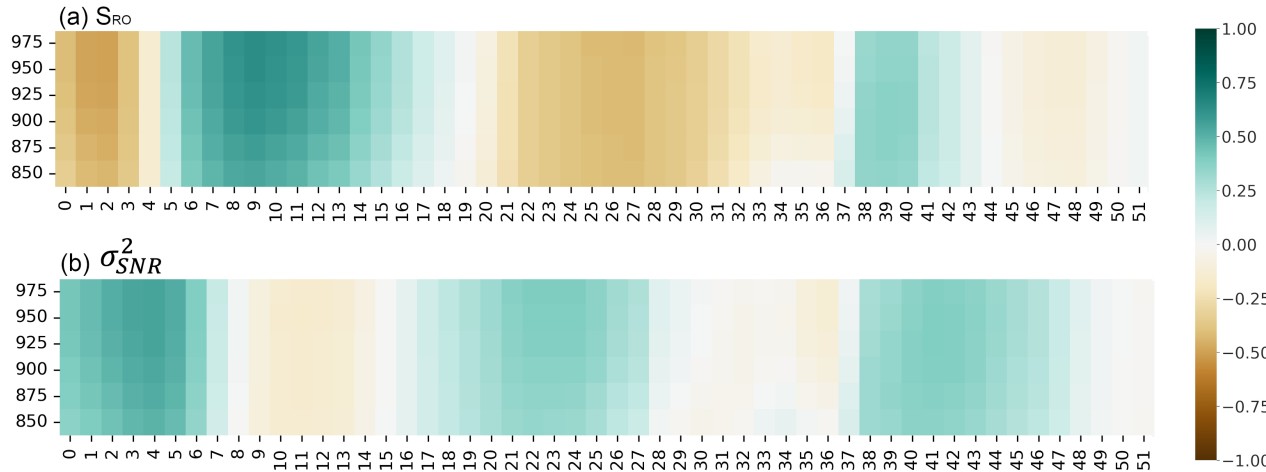

**Figure A2.** Same with Fig. 2, except for Metop-B training dataset.

*Author contributions.* D.L.W. came up with the initial idea. J. G. designed the methodology, executed the plan, built the model, and conducted the validation and data analysis. M.B. helped conducted the hyperparameter grid search. M.G. provided Fig.1. M. Z. provided the high vertical resolution AR-Recon data. All authors participated in result discussion and interpretation.

*Competing interests.* The authors have no competing interests.

*Acknowledgements.* J.G. is grateful to Dr. Mariel Friburg at NASA Goddard in providing financial support of M.B. We thank the editor and two reviewers for their thoroughly and insightful comments, which helped greatly in improving the readability and clarity of this paper.

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
