# Peer review of "A Machine-learning Based Marine Atmosphere Boundary Layer (MABL) Moisture Profile Retrieval Product from GNSS-RO Deep Refraction Signals"

_EGUsphere, 2024_

## Author Comment (AC1)

Reviewer #1:

Response summary:

We thank the two reviewers gratefully for their helpful and constructive suggestions. Please see our responses in **blue letters** below every comment, and **red letters** highlight the changes made in the revised manuscript. Following the reviewers' suggestions, we had made some major updates in the revised manuscript. In particular:

(1) we didn't perform the SNR retrieval for Metop-A and Metop-B in the original submission after observing slightly different non-linear relationships (Fig. A1 and A2) compared to COSMIC-1 (Fig. 2). In the revised manuscript, 2 additional ML models are trained for Metop-A and Metop-B, respectively, and are used for generating predictions (updated Table 1) to enhance the robustness of subsequent climatology and diurnal variation studies. Note that adding in Metop-A and Metop-B retrievals only add 14 additional samples to the collocated SNR-ML retrieval – radiosonde/dropsonde data samples during the MAGIC campaign.

(2) Meanwhile, please accept our apology if there's any confusion in the wording in the original submission. The key purpose of this work is to **prove that SNR can be used to generate profile-by-profile MABL (marine atmosphere boundary layer) specific humidity retrievals, and retrievals generated by the SNR-ML method have comparable or better quality to the operational wetPrf/wetPf2 products and ERA5 reanalysis in different weather regimes globally with 20 – 80% more successful retrievals than wetPf2 products in the lowest MABL.** Although producing a harmonized multi-year program-of-record (PoR) using the current algorithm is our ultimate goal, it is beyond the scope of this current work and we cannot do it in this paper with the limited funding and time.

(3) We added a new Fig. 3 (see also below) to demonstrate the signal coherence at profile-by-profile level so to prove the retrievability using SNR. We can also observe the non-linear response is mission-dependent, which justifies why we need to build 3 individual ML models for each mission series.

(4) We added Section 2.3 to recap the underlying physical mechanisms discussed in Wu et al. (2022) and Fig. 3 to justify the reason of using ML instead of physical models for realizing the operational retrieval with the SNR measurements.

(5) We added a new Fig. 5 to illustrate the model internal architecture and deleted terminologies in the text that might be just jargons to readers.

(6) We redraw Fig. 2, Fig. 4, Fig. 9, Fig. A1 and A2 to improve the readability.

(7) We fixed a code bug in plotting Fig. 11 (previous Fig. 9). We also found a bug that the total number of collocated wetPrf/wetPf2 product is pressure level dependent. Therefore, we added a new Table 3 to reflect the correct collocated sample sizes at

each pressure level for each campaign. The entire Section 3.3 is largely rewritten to reflect the changes of major findings.

[Figure]

Fig. 3 in the revised manuscript: density plots of the specific humidity – SNR relationship constructed from the training dataset for (a) Metop-A; (b) Metop-B and (c) Cosmic-1. The ERA-5 950 hPa specific humidity between 45S and 45N and collocated and coincident SNR values at SLH = -100 km are used to construct these density plots. (d) is a reproduction of Fig. 9c in Wu et al. (2022) to show the same correlation but from monthly gridded data using Cosmic-1. One can easily see the linear relationship that was observed and reported in the Level-3 product in Wu et al. (2022) holds at Level-2 profile-by-profile level, but with much larger noise (hence a ML model would handle it better than a traditional multi-variable regression model).

General comments

This paper proposes a method to obtain vertical profiles of water vapor from GNSS radio occultation (RO) observation in the marine planetary boundary layer (MPBL) using machine learning (ML), which is a form of artificial intelligence (AI). I know little about ML and AI (the paper should be reviewed by an expert in this area), so I reviewed the rest of the paper. The basic idea may be useful in a practical sense, but in my opinion it

is not acceptable for publication because it is difficult to understand and is unclear and imprecise in many places. Thus, it does not make a convincing case of the merit of using ML/AI to improve RO retrievals of water vapor in the MPBL.  I recommend major revisions with care taken to use clear, precise, and understandable language.

We highly appreciate your comments with concerns about whether we are leaping forward too quickly without understanding the true physical mechanism clearly. In fact, the possible physical mechanisms for SNR to contain MPBL water vapor information have been discussed in detail in Wu et al. (2022). Now in the revised manuscript, we add Section 2.3 to summarize them, which includes normal bending (i.e., the situation when L2 retrieval algorithm can converge), grazing reflection (multi-path reflection), super-reflection, ducting or diffraction. SNR signal can be used for retrievals under all these 5 scenarios while the operational WetPrf/WetPf2 retrieval algorithm in the majority cases works only under the first scenario (i.e., normal bending). In reality, complex MABL can produce a mixed effect in the soundings from a combination of these scenarios. As a result, sophisticated physical radiation transfer models (e.g., radiohologram, canonical transform) can surely be used but at the expense of high computational costs and hence impractical operationally. Moreover, the retrieval itself is essentially still an under-constraint problem, which commonly occur for satellite retrievals and assumptions (no matter physically making sense or not) need to be made to fully constrain the physical model. As the quasi-linear relationship is preserved at profile-by-profile level with larger noise compared to the monthly gridded and smoothed data (new Fig. 3), and the height-dependency of the regression coefficient is highly non-linear (Fig. 2), a ML model is simply the best choice to extract the signal.

We'd like to reiterate that the scopes of this work are (1) to provide a practical way to extract MABL water vapor information from the SNR signal (2) demonstrate the feasibility of this method and the science value of it (now Lines 80-82 in the revised manuscript). We thank you very much for helping us rethink the scopes and values of this work and hopefully the revision makes them clear to the readers now.

A clear description of the scientific basis for the method, under what atmospheric conditions it is valid and useful, its limitations, and how it compares with 1D-Var retrievals of water vapor profiles in the MPBL would be useful. This is especially important for this paper, since it presents results from a technique that is unfamiliar to most experts in radio occultation. There are some odd words and phrases that should be replaced with more scientific or precise words. Please see some examples in the detailed comments section; these are only examples; I stopped looking carefully for language issues after a while.

Thanks very much for pointing out the language discrepancies and gave some valuable detailed suggestions below! We've gone through polishing the language much more carefully this round and hopefully clears up jargons/grammar errors that would negatively impact the reading experience.

A recent paper that is relevant to the discussion regarding SNR and the $H_{SL}$ is Sokolovskiy et al. (2024), and this paper should be referenced in the Introduction (somewhere after Line 65).

Thanks for recommending reading this new publication! It is now cited in Section 2.3 when we recap the possible physical mechanisms to explain the information content in SNR.

The paper uses several different names for the ML method, most often "SNR-method" which is misleading. I suggest using the more descriptive name "ML-SNR method" (as done in Line 16) consistently throughout the paper. Whatever acronym is used, it should be consistent everywhere.

Thanks for this nice suggestion! Now we've changed "SNR-method" to "SNR-ML method" consistently throughout the text.

The authors use marine planetary boundary layer (MPBL), which is OK. But they may wish to consider using marine atmospheric boundary layer (MABL) instead to be consistent with what they used before in the closely related paper Wu et al. (2022).

Thanks for pointing this out! We have now changed all MPBL to MABL to keep consistency. When talking about the general PBL, we still keep this terminology considering its wide usage.

The numbering of Section 2 is currently:

2. Data and Model

2.1 Training and Validation Datasets

2.1.1 Machine Learning Model Selection

The number of Section 2.1.1 should be changed to 2.2.

Thanks for catching this mistake!

It would be useful to have a short simple summary of the steps used to train the model and then to validate it, perhaps including a numbered series of steps in the process or a flow chart. This could go at the end of Section 2. The highly technical first paragraph of Section 2.1.1 is not very useful to the nonexpert in ML. A new subsection to to Section 2 could be added which included the series of steps or a flow chart showing the steps in the process: 2.3 Summary of ML-SNR model and validation

Thanks for the suggestion. We have added a Fig. 5 and rewritten the text of the 1st paragraph of Section 2.2 to give a summary of model architecture and input/out parameters, and the rest technical jargons have been removed. We also include the training codes together with the training and validation dataset. Hopefully the rearrangement of the three subsections under Section 2 logically make more sense to the readers now.

The authors use ERA5 MPBL water vapor data to train the ML-SNR model and then test the model using independent datasets. Although ERA5 is a well-tested and widely used reanalysis, there are likely significant uncertainties in the MPBL water vapor analysis, so it is only an approximation to "Truth." A ML model trained on ERA5 data that is tested with an independent data set will return retrievals that are consistent with ERA5. This seems to be the case in Fig. 4, although there is a lot of scatter. The comparisons of the ML-method retrievals of water vapor to radiosondes as done in the paper will contain the influence of ERA5 data. It would be useful to discuss the influence of the training data set on the retrievals. It would also be interesting to discuss how the retrieval of individual water vapor profiles would be used if the scatter (or uncertainty) of each profile is as large as the scatter in Fig. 4 suggests.

We totally agree with you that ERA-5 is not an "ideal truth" dataset to train upon. If we could have enough collocated shipborne radiosonde profiles or other dense ground-based measurements over the global ocean, we'd be more than happy to train against those. However, they do not exist, and ERA-5 is the "best available" to us. This has been discussed in the 3rd paragraph in Section 2.1. Nevertheless, the lower end of Fig. 4 (now Fig. 6 in the revised version) disagrees with ERA-5 significantly, indicating some real information is gained from the SNR signal when ERA-5 is essentially a flat value. Some sentences are added to Section 3.2 "uncertainty quantification" now to emphasize this point.

We also agree with you that some of the radiosonde data have been assimilated to the ERA-5 data. Based on the incomplete knowledge we learnt from the campaign coordinators or data distributors, for the campaigns that NOAA partially or fully funded (e.g., ARRecon, EUREC4A, ATOMIC), the radiosonde data have probably been assimilated or will be assimilated in the near future (so not in the current ERA-5 version

that we use here). There is no clear documentation to trace which are assimilated or not. For the rest campaigns that are sponsored by NSF, DOE or other agencies, the answer is probably no. Therefore they can be considered completely independent validation datasets. It is worth noting that there are several publications that use ARRecon or EUREC4A radiosondes as "independent dataset" to evaluate ERA-5 water vapor biases (e.g., Cobb et al., 2021; Kruger et al., 2022).

Cobb, A., A. Michaelis, S. Iacobellis, F. M. Ralph, and L. Delle Monache (2021): Atmospheric River Sectors: definition and characteristics observed using dropsondes from 2014-20 CalWater and AR Recon, Mon. Wea. Rev., doi:10.1175/MWR-D-20-0177.1

Krüger, K., Schäfler, A., Wirth, M., Weissmann, M., and Craig, G. C.: Vertical structure of the lower-stratospheric moist bias in the ERA5 reanalysis and its connection to mixing processes, Atmos. Chem. Phys., 22, 15559–15577, https://doi.org/10.5194/acp-22-15559-2022, 2022.

One of the intriguing findings is that this SNR retrieval product frequently outperforms the operational L2 wetPf2 product when evaluating against the shipborne campaign radiosonde profiles. Take the ARRecon as an example (Now Fig. 9, comparing the solid orange triangles from SNR method and open orange triangles from wetPf2). For the atmospheric river events, operational L2 retrieval seem to fail less in the PBL (only 20% less samples than using the SNR method) versus other regimes (>100% less collocated samples). However, one can see the SNR retrieval results are closer to the 1:1 line while wetPf2 tend to be dry biased in the atmospheric river events. Although operational wetPf2 product does not rely on ERA-5 reanalysis (albeit still use ECMWF IFS as the initial conditions in some 1DVar algorithms), however, since the temperature and water vapor contribution cannot be independently separated in the refractivity profile, wetPf2 retrievals do not necessarily produce better agreements with ground "truth".

As for your last point, in practice we suggest users to not use retrieval with estimated uncertainty larger than 50% (Section 3.2). In addition, users need to be cautious to use retrieval in the deep tropics because the underlying physical explanation is frequently violated. These suggestions are included in the abstract and the conclusion. It is impossible to quantify how much added value or independent information come from SNR vs. from the training dataset from ERA-5. Nevertheless, the across-board better performance of SNR method versus wetPf2 shown in Fig. 11 statistical correlation suggests we gain scientifically valuable information from real deep-SNR measurements that the training dataset does not contain.

We have included some of the response above in the last paragraph of Section 3.3.

"To summarize the major findings for comparisons against the limited independent comparison against radiosonde/dropsonde datasets available over the open ocean, we can draw the following conclusions. Firstly, SNR-ML retrieval are the best at capturing the MABL vertical structure from the surface (975 hPa) up to 850 hPa globally compared to the ERA-5 reanalysis except at the deep tropics. The reason for the poor performance of the SNR-ML method in the deep tropics is probably due to the breakdown of the original assumption: turbulence and mixing in the tropical MABL by frequent shallow convections constantly disrupt the ducting condition, causing SNR reemerging at the deep HSL blending other information and hence are not useful for MABL water vapor retrieval. Secondly, compared to the operational Level-2 retrievals, the SNR-ML method can achieve 20 − 80% more samples in the MABL, especially over high-latitudes or in the deep tropics. Although some of the "independent validation dataset" is not completely independent as they may have been assimilated in the ERA5, the fact that SNR-ML retrieval statistics outperform ER-5 at all 6 pressure levels in diverse weather regimes prove that real physical information from SNR observations is learnt and kept by the ML model for prediction, admittedly it is impossible to quantify how much the real observed information contributes without accurate physics-based modeling simulations."

The physical basis for the correlation of SNR as a function of the HSL with MPBL water vapor content, which was found by a related paper Wu et al. 2022 (I did not review this paper), is not explained well. The availability of meaningful SNR (SNR above the noise level) at all HSL levels depends on the boundary layer structure. For moist boundary layers that have no sharp inversions, this correlation is understandable; usable SNR are available all HSL levels in their model. However, for dry boundary layers there may be no useful SNR at deep levels, and for moist boundary layers with sharp inversions, there may be HSL levels with no useful values of SNR. The difference between dry and moist boundary layers and moist boundary layers with and without sharp inversions, and the effect of ducting and superrefraction should be discussed. All boundary layer structures are lumped together in this paper. Related to this issue is the confusing sentence beginning in Line 68 "The paper attributed such a positive correlation to the strong refraction from a horizontally stratiform and dynamically quiet MPBL water vapor layer that acts to enhance the SNR amplitude at deep $H_{SL}$ through ducting and diffraction interference." A similar issue exists with the sentence in Lines 244-247, which I do not understand.

We've now added Section 2.3 to discuss the several physical conditions that allow deepSNR to carry real physical information. All those conditions require certain level of "flat surfaces" for the radio waves, which include both a sharp and smooth PBL top and smooth surface. When convection happens, both PBL top and surface (due to strong wind) are not smooth anymore.

The paper refers to the 1D-Var retrieval of water vapor as "the standard Level-2 product" (line 12) which is imprecise and will mean nothing to most readers. Apparently it refers to the retrieval of water vapor and temperature from 1D-variational analysis (wetPrf in their paper). Please use a clear and precise term for this product and define it.

I see what you mean. Instead of explaining the details how L2 algorithm works, we now modify the 4th paragraph in Section 1 when Level-2 product is first introduced. It now reads as "GNSS Radio Occultation (GNSS-RO) retrieves temperature and water vapor profiles using the 1D-Var approach routinely from the Level 2 bending angle product (referred as "standard L2 product" or "operational L2 product" hereafter), the latter of which is used operationally in numerical weather data assimilation systems to improve weather forecasts (e.g.,Kuo et al. (2000)).".

Johnston et al. (2021) use the newer wetPf2 water vapor data (Wee et al. 2022.) The Gong et al. paper refers to wetPrf and uses it in the comparison with the ML-SNR retrievals. WetPrf is the older and less accurate retrieval.

Sorry that I'm not fully aware the difference between wetPrf and wetPf2. We read this Wee et al. (2022) paper and double checked the UCAR website. The WetPf2 retrievals are used after 2016 for COSMIC-1 and COSMIC-2 satellite missions, while the 2012 and 2013 comparison still used WetPrf data because WetPf2 is not available for download. For Metop-A and Metop-B, we couldn't find wetPf2 product on the UCAR website, and hence we downloaded the wetPrf retrievals for UMD web site. We have clarified and cited this paper when operational product is introduced in Section 2.1 when introducing the training and validation datasets. To keep consistency, we updated Fig. 1 COSMIC-1 curve with WetPf2 now. The new added paragraph in Section 2.1 reads as:

"GNSS-RO operational water vapor retrieval product provided by the University Corporation for Atmospheric Research (UCAR) is employed to evaluate the quality of the SNR-ML retrievals. This operational product is called "wetPrf" for data collected before 2013, and "wetPf2" afterwards. The latter has better penetration depth (Wee et al. (2022)) and is used for constructing Fig. 1, but "wetPrf" product is used for the MAGIC campaign comparison because it was carried out during 2012 - 2013. Note that the key Level-2 profile to enable the 1D-VAR retrieval used by the wetPrf/wetPf2 product is the bending angle, which is assimilated in the ERA-5 reanalysis. Therefore, this is not an independent evaluation dataset. The purpose of this comparison is to identify the merits and caveats of the SNR-ML retrievals against an existing mature product."

Why are the differences in the penetration rates in atmPrf and wetPrf different in COSMIC-1 (blue) and COSMIC-2 (red)? In COSMIC-1 the wetPrf retrieval rate at low levels is greater than the atmPrf retrieval rate (e.g. at 4 km, ~28% for wetPrf and less than 2% for AtmPrf). In COSMIC-2, the opposite is shown; the penetration rate for atmPrf is greater than that for wetPrf in the low levels. Why do the authors use the number of Level-1B files as the denominator; it would be better to use the total number of each files (profiles) in the denominator (success rate = number of retrieved values/number of profiles).

After some inspection, we found that we used wetPrf for COSMIC-1 (version 2013.3520) and wetPf2 for COSMIC-2. That is probably the reason why the success rate is different for atmPrf and wetPrf. The figures now have been replaced with and atmPrf and wetPf2 for COSMIC-1 from an up-to-date version (2021.0390). The descriptions of success rate in the main text are updated accordingly. There is only very minor difference between wetPrf and wetPf2 for the COSMIC-1 success rate (30% ->40% at 0.5 km, 60% -> 55% at 1 km), but atmPrf seems to be more sensitive to version changes. This could be due to advances in excess phase computations, retrieval software, GNSS orbits, clock, and earth orientation products (UCAR Data Release, 2022).

[Figure]

Old Fig. 1 (left two panels) vs. new Fig. 1 (right two panels; updated with 2021.0390 version).

How are the uncertainty values in Section 3.2 and Fig. 6 defined and determined?

Please refer to the new Fig. 5 for the ML model internal architecture. In each layer, we randomly drop 25% of the neurons during training. Since it's random dropping, the neuron nodes could be critical parameters or only remotely related. Through the 100 epochs of training, the dropping process can partially capture the stochastic nature of the relationship between inputs and outputs. Then when we run prediction for the independent validation dataset, we run 30 predictions for each input sets. The standard deviation of the 30 predictions is used as the prediction uncertainty. This is a widely acknowledge method in the ML community to estimate model uncertainty (Gal and Ghahramani, 2014). Although we found that in real world retrievals, this method

tends to under-estimate the real uncertainty, it is well correlated with the uncertainty generated from physical algorithms (in a completely different retrieval project, so the conclusion might not be generalizable, but we are working on a paper to report this).

Yarin Gal and Zoubin Ghahramani: Dropout as a Bayesian Approximation: Representing Model Uncertainty in Deep Learning, Proceedings of the 33 rd International Conference on Machine Learning, https://proceedings.mlr.press/v48/gal16.pdf

The References are not in alphabetical order.

Thanks! We have corrected this.

Lines 55-56-In addition to decreasing SNR, which limits the vertical penetration of the RO profiles, superrefraction in the PBL is an issue. Superrefraction makes it impossible to obtain a unique bending angle profile.

In Wu et al. (2022) paper, super-refraction is mentioned as one of the possible mechanisms to cause L2 retrieval failure and let reemerged SNR carry MABL information. Now this mechanism is further elaborated in the new Section 2.3 which lays out several possible physical mechanisms.

Figure 2 needs to be improved. The numbers on the x- and y-axis are not legible, and the axes are not labeled. It appears that there are two figures in 2a and 2b, grid indices at the top and correlations with ERA5 specific humidities in the lower right corner. But the lower right corner is solid dark green, indicating a perfect correlation of 1.0? The other "boxes" at the bottom of the figure to the left of the solid green box at the lower right corner oscillate between positive and negative correlations, and this should be discussed,

After secondary thoughts, we believe the cross-correlations among different excess phase levels are less important to show than to demonstrate that all levels are correlated with MABL specific humidity, which justifies the reason why we'd like to use the entire profile instead of just one level for the retrieval. However, we agree that this figure is too busy and hard for readers to capture the main idea. They are now cropped to only show the bottom row (see below). The other two figures in the appendix are also updated. Specific humidity values are highly correlated among different pressure levels but are not exactly 1:1 correlated. The previous cross-correlation figure is too crowded to differentiate the gradient.

[Figure]

Figure 2. Correlation between collocated ERA5 specific humidity at 975 - 850 hPa and SRO (top) and σ2 SN R (bottom) at various excessive phase levels from the training COSMIC1-ERA5 dataset). Only grid indices are shown in the axis titles, and the corresponding Log10(φL1) values can be found in Table B1.

The caption refers to Table A2, which does not exist.

Thanks a lot for spotting A2. It's a wrong reference label when editing the Latex doc and has been fixed.

Fig. 3 has a lot of blank space and in the three regions of campaigns it difficult to see the details. Consider three separate maps of the three regions.

Thanks for the suggestion! Now three regional maps are included in the new Fig. 4.

[Figure]

Figure 4. Maps for radiosonde/dropsonde locations from different shipborne or airborne campaigns in (a) tropics; (b) mid-latitudes; (c) southern ocean. Detailed campaign information can be found in Table 2.

Figure 7 also needs a better explanation. The gray dashed line (ERA5) is difficult to see. How are the solid black lines and the dashed black lines constructed? They are irregular

so they don't look like best fit lines. I presume the solid thin straight black line is the 1:1 line, but why does it not extend to the corners of the grid? SNR retrieval should be ML-SNR retrieval in the caption. "Level-2 retrievals" in the caption should be "wetPrf" retrievals. There are very faint orthogonal red and black lines in the figure—what are these? This figure contains a lot of information and detail; consider breaking into two figures and/or making them larger. Presenting results at 4 pressure levels rather than 6 might help. This is an important figure and should be clear and explained well.

Thanks for your detailed suggestion! We have now updated Fig. 7 (now Fig. 9) with adding in the new Metop-A and Metop-B collocations, and rewritten the figure caption to clarify some misunderstandings. We cannot make these panels less busy because the among of information to deliver is indeed a lot. The ERA-5 line is replaced from grey to brown now in color. Fig. 9 and Fig. 10 are both huge in image size (3162 X 2475 pixels with 330 dpi), so readers interested in scrutinizing details can enlarge the image up to 200% of the original size and still see everything clearly.

To clarify the meaning of the lines along the diagonal direction: the thin straight lines are the 1:1 line for reference. Black bold solid lines are made by connecting the mean values of each campaign (now changed to big grey symbols with errorbar), so do the black dashed lines (L2 operational product) and grey dash-dot lines (ERA5 subsamples). Now the caption for Fig. 9 reads as:

Figure 9. Scatter plots of collocated specific humidity comparison between radiosonde "truth" and retrievals from SNR (closed symbols) and Level-2 standard retrieval (open symbols) for each pressure level. Black thin diagonal lines are the 1:1 lines for reference. The mean and standard deviation from the SNR-ML retrieval from each campaign are shown as bigger symbols with the same color. In addition, the mean retrieved values from each campaign as opposed to the mean from radiosonde "truth" are shown as the bold black lines for SNR-ML retrievals, bold black dash-dotted lines for wetPf2 retrievals, and grey dashed lines for ERA-5 from the subset where collocations are found for SNR-ML and radiosonde data samples.

I did not review Section 4 carefully.

Detailed comments

1. Lines 1 and 3, also Line 299—what kind of gradient? Horizontal gradient or vertical gradient?

   Vertical gradients. Words added now.

2. Line 5 Define SNR as signal to noise ratio—it is not an acronym for deep refraction signals.

   Now changed from "signal" to "signal-to-noise ratio (SNR) signal".

3. Line 7—what is "pixel-level water vapor profiling?" What pixels?

   Now changed to "profile-by-profile water vapor retrieval".

4. Lines 30-31: What is "polar proneness to the climate change"?

   What I mean is polar region is more prone to climate change than other places because of the positive reinforcement of temperature-ice melting feedback. The wording is now changed to "proneness of polar area to the climate change".

5. Line 38: grammatically incorrect; I suggest rewriting "Emissions from clouds often overwhelm the emission signal…"

   Thanks! Grammar error corrected.

6. Line 39: Delete "in the scene"

   Thanks! Deleted.

7. Line 44: delete "which couldn't be used to gain….MPBL."

   Now changed to "which still lacks the vertical information of WV in the MABL".

8. Line 46-use "high resolution" rather than "superb resolution" and give the nominal vertical resolution (100-200 m).

   Thanks. Changed.

9. Line 52: "coarse horizontal resolution" should be replace by "relatively large horizontal footprint." Resolution refers to the average distance between observation points, footprint refers to the spatial scale of the atmosphere that affects the observation (see Boukabara et al. 2021).

   Well I would disagree with this statement. For nadir-looking, cross-track scan or conical scan instruments, footprint is defined as the half power beam width (HPBW), which is determined by the satellite antenna size and distance from Earth's surface. For limb sensors like GPS, the horizontal resolution is determined

by the weighting function width and viewing geometry. "Footprint" is not a standard terminology to use for limb sensors.

10. Line 53-typo-concern.

    Thanks! Corrected to "concern".

11. Line 53- What does the sentence "This is typically not a big concern in MPBL as vertical gradient if much sharper and harder to characterize if not using in-situ measurements )e.g. shipborne radiosonde" mean?

    Now changed to "This is typically not a big concern in MABL as vertical gradient is much sharper than horizontal gradient and harder to characterize  " What we mean is MABL vertical gradient is sharper than horizontal gradient usually, so coarse horizontal resolution using GNSS-RO is not a big concern considering its relatively good vertical resolution.

12. Line 56—SNR decreases with decreasing height. Current sentence "decreases with height" implies that it decreases upward.

    You are absolutely correct. The sentence now has been rewritten as "decreases with decreasing altitude".

13. Lines 57-59—This is misleading. It says the water vapor retrievals fail to converge because they require a high SNR, when in fact the main issue is that the RO signals do not penetrate deeply enough because of decreasing SNR near the surface.

    Yes, we agree with you. Now the sentence is modified as "the Level-2 radio-occultation (RO) signal hence often does not meet the SNR threshold near the surface.  As a result

     the GNSS-RO 1D-Var based retrievals fail in the MABL due to weak RO signal."

14. Line 61—Fig. 1 does not show that C2 has improved its SNR—it shows that C2 has a deeper penetration rate that C1, which is a result of higher SNR. This is another example of an imprecise/incorrect statement.

    Please see response above.

15. Line 64 and 119-The Maneshan et al. (2024) reference is not in the References section.

    This paper was just submitted to AMT when the current manuscript was submitted the same time. Now Maneshan et al. (2024) is published on AMTD. We have updated the citation.

16. In Fig. 1, it should be stated that the height is mean-sea level rather than impact height, which is often used, and the y-axis should be labeled MSL (km).

    Caption of Fig. 1 is changed to clarify it's "height above sea-level". The wetPrf data has been replaced with wetPf2 for both COSMIC-1 and COSMIC-2 for consistency, and the pressure level is added to the right axis.

17. Line 72-delete "understandably"

    Done

18. Lines 79-80: These two sentences could be replaced by something like "Artificial Intelligence/Machine Learning (AI/ML) has been increasingly used in remote sensing in recent years."

    Done.

19. Line 95—delete "thoroughly"

Done.

20. Line 110-define $f_{L1}$

Phi_L1 is L1 excess phase.

21. Line 112: How can you say ERA-5 is the best reanalysis? I agree that it is very good, but the best? Do you mean ERA5 is better than MERRA-2 in the metric you talk about in the next few sentences?

    Yes. In terms of PBL specific humidity bias, ERA-5 is less biased than MERRA-2 in all latitudes according to Johnston et al. (2021). For many other variables related to cloud and precipitation, that's the same case but since it's irrelevant, the meaning of this statement should be confined and not be extrapolated. I think the first sentence in this paragraph is clear about the condition for this statement. Do you feel it's not strong and clear enough?

Lines 113-114—Johnston et al. (2021) used the improved wetPf2, not wetPrf.

Corrected.

22. Line 144---Reference for CNN model?

Citation LeCun et al. (2015) is now added.

23. Line 144 and 157—replace "old-fashioned" with something like "earlier" or "simpler".

Changed to "earlier". CNN is revolutionary but not necessarily better than some simple ML models in all use cases. Our project is a rather simple case for ML, so performance is quite comparable with using those "earlier" ML models.

24. Lines 202, 235 and other places—Use "wetPrf" retrievals instead of "Level-2 retrievals."

Done. "Level-2" had been either deleted or replaced with "wetPrf/wetPf2" or "operational".

25. Line 258---Rewrite to say "...the general patterns in the ML-SNR method and MERRA-2 specific humidities agree fairly well."

Reword to "The geographic distribution of highs and lows and their gradients are in general agreeable."

26. Line 263---I would not describe the comparisons shown in Fig. 11 as "more boring." I am not sure what is meant by this characterization. Perhaps it means that the structures are less complicated than in Fig. 10, or that the agreement is better? In any case, that is not necessarily "more boring."

Relaced "more boring" with "lacking geographical variations".

27. Line 270—delete "notorious."

Done.

28. 12 is not mentioned in the text of section 4.2. It should be introduced in the text somewhere around Line 275.

Oh thanks for spotting this stupid overlook!

29. Line 287—replace "drops down" with "decreases."

    Done.

30. Line 293—delete "from this exercise."

    Done.

31. Line 296—replace "will keep this topic foggy" by something like "make observing and verifying the true diurnal cycle difficult."

    Done.

32. Line 297—delete "disentangle this mystery"

    Deleted.

References

Boukabara, S.-A., J. Eyre, R.A. Anthes, K. Holmlund, K. St. Germain, and R.N. Hoffman, 2021: The Earth-Observing Satellite Constellation: A review from a meteorological perspective of a complex, interconnected global system with extensive applications, *IEEE Geoscience and Remote Sensing Magazine,* **9**, 3, 26-42. https://doi.org/10.1109/MGRS.2021.3070248

Sokolovskiy, S., Z. Zeng, D. Hunt, J.-P. Weiss, J. Braun, W. Schreiner, R. Anthes, Y.-H. Kuo, H. Zhang, D. Lenschow, and T. VanHove, 2024: Detection of super-refraction at the top of the atmospheric boundary layer from COSMIC-2 radio occultations. *J. Atmos. and Ocean Tech*., **40**, 65-78.  https://doi.org/10.1175/JTECH-D-22-0100.1

Wee, T.-K.; R.A. Anthes, D.C. Hunt, W.S. Schreiner, and Y.-H. Kuo, 2022: Atmospheric GNSS RO 1D-Var in Use at UCAR: Description and Validation. *Remote Sens*., **14**, 5614. https://doi.org/10.3390/rs14215614

Thanks much for sharing these recent relevant publications which we overlooked. They have now been included in the reference list and in appropriate locations in the main text as citations.

---

## Author Comment (AC2)

Reviewer #2:

Summary:

We thank the two reviewers gratefully for their helpful and constructive suggestions. Please see our responses in **blue letters** below every comment, and **red letters** highlight the changes made in the revised manuscript. Following the reviewers' suggestions, we had made some major updates in the revised manuscript. In particular:

(1) we didn't perform the SNR retrieval for Metop-A and Metop-B in the original submission after observing slightly different non-linear relationships (Fig. A1 and A2) compared to COSMIC-1 (Fig. 2). In the revised manuscript, 2 additional ML models are trained for Metop-A and Metop-B, respectively, and are used for generating predictions (updated Table 1) to enhance the robustness of subsequent climatology and diurnal variation studies. Note that adding in Metop-A and Metop-B retrievals only add 14 additional samples to the collocated SNR-ML retrieval – radiosonde/dropsonde data samples during the MAGIC campaign.

(2) Meanwhile, please accept our apology if there's any confusion in the wording in the original submission. The key purpose of this work is to **prove that SNR can be used to generate profile-by-profile MABL (marine atmosphere boundary layer) specific humidity retrievals, and retrievals generated by the SNR-ML method have comparable or better quality to the operational wetPrf/wetPf2 products and ERA5 reanalysis in different weather regimes globally with 20 – 80% more successful retrievals than wetPf2 products in the lowest MABL.** Although producing a harmonized multi-year program-of-record (PoR) using the current algorithm is our ultimate goal, it is beyond the scope of this current work and we cannot do it in this paper with the limited funding and time.

(3) We added a new Fig. 3 to demonstrate the signal coherence at profile-by-profile level so to prove the retrievability using SNR. We can also observe the non-linear response is mission-dependent, which justifies why we need to build 3 individual ML models for each mission series.

(4) We added Section 2.3 to recap the underlying physical mechanisms discussed in Wu et al. (2022) and Fig. 3 to justify the reason of using ML instead of physical models for realizing the operational retrieval with the SNR measurements.

(5) We added a new Fig. 5 to illustrate the model internal architecture and deleted terminologies in the text that might be just jargons to readers.

(6) We redraw Fig. 2, Fig. 4, Fig. 9, Fig. A1 and A2 to improve the readability.

(7) We fixed a code bug in plotting Fig. 11 (previous Fig. 9). We also found a bug that the total number of collocated wetPrf/wetPf2 product is pressure level dependent. Therefore, we added a new Table 3 to reflect the correct collocated sample sizes at each pressure level for each campaign. The entire Section 3.3 is largely rewritten to reflect the changes of major findings.

The manuscript presents an interesting idea to harness other available information of a radio occultation observation to improve the retrieval of water vapour in the lowest boundary layer (marine here). I do have though major concerns regarding how data is selected, presented, and conclusions drawn. It is unclear whether the different retrieval / re-analysis comparisons are actually using the same data, or whether different data is entering each retrieval, and thus comparisons are misleading. I'd also like to see or at least discussed, whether a more advanced retrieval setup can be used, that uses SNR/amplitude and bending angles to improve the data in the lowermost atmosphere. A physical based retrieval will certainly be preferred by the NWP and Climate community, rather than one that needs per satellite tuning. There is also no real discussion on the higher SNR available with e.g., COSMIC2, and whether that improves the SNR retrieval (in fact, it is actually fairly unclear where C1 and where C2 is used, except that C2 is not available at higher latitudes and for certain periods). Metop is only included in the appendix, and it is unclear why that data wasn't used as well.

Thanks for your constructive comments. We apologize for unclarity in many places. Now as also suggested by Reviewer #1, we trained two additional ML models for Metop-A and Metop-B respectively. The periods and mission names for training, independent testing and prediction are listed in Table 1. The evaluation in Section 3.1 and 3.2 used the "testing periods" following standard ML procedures, while from Section 3.3 and onwards, data in the "prediction periods" are fed into the computation.

We didn't train a separate ML model for COSMIC-2, but rather apply the ML model trained solely on COSMIC-1 data to the COSMIC-2 SNR profiles for prediction. Then we used collocated radiosonde and the COSMIC-2 predictions during the 2020 ARRecon, EUREC4A and ATOMIC campaigns for validation. This is to test the robustness of the so-called "transferred learning", which seems to be good based on the consistent performance. However, we had to sacrifice the higher-frequency sampling rate for COSMIC-2 and to down-sample the SNR data to 1 Hz in order to use the COSMIC-1 ML model. In the future we might want to develop one ML model for one mission.

Regarding the joint bending angle + SNR retrieval, this is a great idea. I would picture it being especially useful when refractive index is negative and where the bending angle is biased (Feng et al., 2020). They can complement each other in different PBL scenarios. However, how to realize that in practice has a long way to go. At least for

now, we can see two major issues: (1) bending angle is in height (or pressure) coordinate, while deepSNR signal is below surface so it is difficult or maybe impossible to mirror the signal to a specific above-surface coordinate as the deepSNR signal probably comes from multi-paths (a toy model is given in the appendix of Wu et al., 2020). (2) as now discussed in Section 2.3, the underlying physical mechanisms for deepSNR to carry PBL information are not fully understood. Multiple scenarios can co-exist to cause the information to reemerge from below surface. As a physical deepSNR model is not readily available, we do not see this product to be assimilated in the NWP in the near future,  but can be rather used as another observational reference to evaluate NWP forecasts. This discussion is now included in the conclusion part as outlook to the usefulness of this new product.

One thing to clarify is that "bending angle" is a level-2 product from GNSS-RO. The satellite-dependent "tuning" (a.k.a., calibration) happens behind the scenes during Level-1 to Level-2 data processing, and many "bending angle" profiles have to be excluded during "quality control" step when its SNR is below certain threshold. However, for these profiles, although "bending angle" cannot be used anymore, the SNR signal reemerged from below surface still contain the physical PBL information. If the SNR profile could be assimilated to NWP models in the future, they would be processed at the front-end and NWP centers do not need to worry about satellite-dependent tuning.

Text based comments (with Page/Line):

P1L10: Is this Monte Carlo anywhere used in the manuscript?

 Yes. The dropout method's full name is "Monte Carlo dropout method". We didn't realize this is an unfamiliar terminology to some readers, and now have modified words in Section 2.2 "Machine learning model selection" to clarify. It now reads as:

"In the prediction step, 30 predictions were carried out given each input set of variables, the mean and standard deviation of which were used as the final prediction and errorbar. It is worth highlighting that in each convolutional and fully-connected layer, a dropout rate of 0.25 is applied to generate the variation, which is then used to calculate the standard deviation of the "ensemble prediction" as a way to measure the retrieval uncertainty. This so-called "Monte Carlo" dropout method was  designed in ML as a standard technique to regularize model over-fitting (Srivastava et al. (2013)), but were also employed widely as a Bayesian-approximation to quantify model uncertainties (Gal and Ghahramani (2016)). Admittedly the current method only provides a quantification for ML model errors. There is no consideration of SNR measurement errors nor propagation of the error to

the final retrievals at this moment, although this is certainly some procedure to be in place in the future works."

Figure 1: Maybe add the approximate hPa as well?

Done. Added.

P5L24: The table referred to seems to be B1. And, are the entries in B1 the right way round? And is excessive phase the same as excess phase (which is commonly used in RO)?

Table B1 in this paper is an updated version of Table A1 in Wu et al. (2022). Thanks for spotting this cross-referencing error. And we have corrected the misspelling. It should be "excess phase".

P5L114: Using RO data to validate reanalysis performance in the PBL and citing it here is misleading on the RO capabilities. Even Johnston stated in the abstract: "Negative C2 moisture biases are evident within the boundary layer, so we focused on levels above the boundary layer in this study.".

Yes. This is a great point. Indeed this is a caveat of using GPS-RO data especially in marine stratocumulus region where negative refractivity index is found frequently and causing bending angle biases (e.g., Feng et al., 2020). However, since there is no absolute "best global truth" observation available for MABL moisture vertical structure, we stretched the result shown in Fig. 3 of Johnston et al. (2021) paper in the boundary layer as an indication of possible systematic biases of humidity in MABL in ERA5 data. In fact, some other studies (e.g., Virman et al., 2021) suggested the opposite. Based on our radiosonde comparison shown in Fig. 10 (previous Fig. 8), there seems to be a slight but systematic dry bias in ERA5 in the MABL at all 6 pressure levels. Nevertheless, since some of these campaigns' radiosonde data are likely having been assimilated in ERA-5, it is nearly impossible to identify some independent dataset to evaluate ERA-5 biases.

Coming back to your point here, we acknowledge our citation was too stretchy, and now add a new sentence at the end of this paragraph, read as: "However, it is also warned in Johnston et al. (2021) that GPS-RO retrievals tend to have its own biases especially in MABL, and in fact some other research suggested wet biases in certain regions (e.g., Virman et al. (2021))."

P5L121: Is there any further restriction in the used C1, C2 data set? Not all occultations / SNR values will go down to the required hPa levels.

For C2 dataset, we took all wetPf2 retrievals (wetPrf for 2012 and 2013) provided on the UCAR data portal and performed the collocation. Therefore, the bending angle profiles that do not pass the SNR threshold should have been filtered out before C2 product is processed (Wee et al., 2021).

For C1 dataset, we didn't describe in detail about our quality control procedures. Now it's added in: "In practice, we use averaged SNR between 35 and 65 km altitude range as the SNR0, and any profile with SNR_0 < 200 or $\sigma^2_{SNR0}$ > 0.05 is considered "low-signal" and is filtered out." ... "In practice, we also filtered out bad open-loop profiles, profiles with data gap greater than 2 km, and profiles with outlier $S_{RO}$ or $\sigma^2_S$ values."

Table 1: What is the validation period? And why not name the prediction period too in the title? And is there any reason for this limited data set use?

The "testing" period is the validation period, and "prediction" is now added to the table caption. There's a terminology confusion between ML field and geoscience field. In our geoscience field, "validation" means the "independent testing" samples in ML terminology, and the "validation" in ML field means the samples used during the training (here it's the randomly picked 10% samples out of the entire training sample).

Downloading hourly ERA-5 reanalysis was extremely slow when we carried out this research, so the training period was limited. Having said so, we got 211,425 training samples for COSMIC-1 data, which we believe is large enough for such an easy job (Fig. 3c). The Metop-A and Metop-B training datasets are relatively small due to the time constraints of the revision deadline. As this is a research work, we feel it is a good starting point to prove the feasibility and merits of this product. Making it operational certainly takes much more effort if it is going to happen in the future.

Figure 2, Caption: The figure does not only show grid levels, but also hPa ones.

Please see the new Fig. 2. Non-useful cross-correlation information has been removed. We hope this new figure is easier to read and more concise in delivering the information content.

P6L125: Is there actually any correlation visible between your 6 pressure levels and latitude? Some of these levels will be always above the PBL, e.g. at higher latitudes with low PBL heights. And is that impacting the retrieval quality, as fewer pressure levels are contributing? Is there an improvement possible; the low water vapor at high latitudes will also lead to limited bending, further complicating the retrieval?

That's exactly why we chose to use a ML method, and include latitude as one of the input variables, because we found the SNR-humidity correlation is the highest at HSL =

-100 km at tropics and mid-latitude, but at HSL = -80 km at high latitudes. Below is the top 10 most important parameters that contributes to the 950 hPa retrieval when we employed the random forest model and the gradient boosting model (those models have to make one prediction at one time):

[Figure]

P6L126: Aren't other levels right above also correlated with the same magnitude?

 Please see the new Fig. 2 for a closer look of the details. We can certainly extend to another pressure level above, but based on independent shipborne radiosonde data comparison (Fig. 9), we can see the performance of SNR-ML method degrades at 900 hPa or above. This is reasonable as SNR-ML should work the best when the PBL has a very sharp gradient if our physical explanation of the working mechanism is correct (Section 2.3).

P7L132: One of the highly relevant advantages of RO data is the independence on the instrument. Having an instrument dependent contribution here will limit any data use for e.g. climate significantly. And, by the way, how does COSMIC-2 look like? And is there a constellation dependent factor too, as C2 observes GLONASS?

Please see the detailed response to your general concerns at the beginning of this response letter. The reason why users see RO data to be independent of instrument is because the calibration has been done for each instrument when processing the received data. The SNR thresholds for filtering out bad RO data are also instrument dependent (Ganeshan et al., 2024).

For the SNR-ML method, it works the same way. We have published the input training data on zenodo, which has been tuned for each individual mission, and interpolated to the excess phase coordinate so to homogenize the signal across different missions (see Fig. 5 in Wu et al., 2022 for an example of its importance). Yet, the non-linear correlation pattern is still instrument-dependent, but COSMIC-1 and COSMIC-2 look nearly identical in this correlation pattern (Fig. 2), and METOP-A and METOP-B look nearly identical in this correlation pattern too (Fig. A1 and Fig. A2).

P7L136: Are all these radio- and dropsondes comparable in their accuracy/sensitivity?

This is a great question but I'm afraid we couldn't provide a knowledgeable answer. Please refer to the citations in Table 2 if you are interested in details of the data quality evaluation for each campaign.

Figure 3: add that the legend numbers are the total number of sondes (I assume this) in campaign.

Added in the caption. And Fig. 3 (now Fig. 4) had been replotted into three sub-panels to show the sonde locations more clearly.

P9L171: "Fig 4, but were otherwise look..." Not sure what this means. Maybe without the "were"?

We meant to say the correlation patterns look extremely similar if we break down Fig. 4 (now Fig. 6) into 6 panels for each pressure level separately.

P9/L176: Ducting would lead to a disappearance of the signal, as it is bended towards the surface. So you need to have strong gradients, maybe getting close to ducting conditions (albeit I doubt that, as we do see signals below -100km regularly, and even further than -200km, likely caused by close to ducting conditions).

We believe ducting could be one of the several causes of the retrievability of MABL moisture information using reemerged SNR signal (Sokolovskiy et al., 2014). A schematics is shown below for the ducting condition:

[Figure]

Fig. 3 from Wu et al. (2022), showing GNSS-RO wave optics as refracted by an atmospheric layer and reflected by the surface in multi-path interference.

P9L182: So you are primarily using only GPS, from COSMIC1 here? What about COSMIC2, and the GLONASS constellation it observes? And maybe make the naming consistent, either COSMIC-1 or COSMIC1.

Here (Section 3.1) includes the standard ML independent testing. Since COSMIC-2 data were not used for training, we shouldn't use that for independent testing per ML standard procedure. The ML model trained on COSMIC-1 data is then applied to the COSMIC-2 SNR profiles to test the robustness of transferred learning (which is robust based on independent radiosonde validation).

Thanks for spotting the inconsistency of abbreviations. We have now use COSMIC-1 and COSMIC-2 throughout the manuscript for consistency.

Figure 4: Suggest to add correlation coeff./std dev to plot.

Added.

Figure 5 a,c: (1) why are these going up to about 22g/kg, and Figure 4 only up to about 18g/kg? Is there some filtering on-going? (2) Keep using the same units, either g/kg or kg/kg in the manuscript.

Thanks for spotting this! During data pre-processing for ML training, every variable is normalized, and we divided every specific humidity value by 22 g/kg to make sure the input values are between 0 – 1 because the largest value among all data points is 22 g/kg. Since Fig. 4 and Fig. 5 (now Fig. 6 and Fig. 7) uses the 3 months of 2018 data for independent testing, the original plotting code for Fig. 4 uses "$q_{max}$" as the axis boundary, and the largest specific humidity value for these three months are 17.8 g/kg, so that's why Fig. 4 (now Fig. 6) has a different axis range. We now have updated Fig. 4

(now Fig. 6) with the same axis range with Fig. 7 (previous Fig. 5), and changed the colorbar units to g/kg in Fig. 7 (previous Fig. 5) to be consistent with other figures.

P11L193: Estimating the SNR contributions to the total uncertainty should be further elaborated, e.g. what contributors do you expect? Horizontal inhomogeneity, ducting, reflections, etc?

This is a very sharp question. The answer is we don't know what exact physical mechanisms cause the uncertainty, and by how much. As summarized in Section 2.3, a couple of atmospheric conditions could co-exist to enable deepSNR retrievals, which we don't have a good physical model to simulate so far. However, as a common sense, we do know when the signal is very weak (i.e., small SNR values), it is hard for any models (physical or machine learning) to separate the signal from the background noise. Hence, retrievals based on small SNR values should come along with a relatively large uncertainty. This is what Fig. 6 (now Fig. 8) and related text tells us.

Figure 6: Please mention the percentage you remove with the 50% uncertainty filter. Maybe also include some further info here on the distribution, e.g. percentage of data below 20% uncertainty?

There are only 2.23% of independent testing data for COSMIC-1 that has uncertainty beyond 50%. If we use 20% uncertainty threshold, about 16.14% retrievals do not pass the uncertainty filter.

P12L209: Please be consistent, if you use only GPS, then don't use GNSS sometimes. 5 lines below, you use GPS again.

We should use GNSS because we use both GPS and GLONASS. The inconsistency has been corrected.

P12L221: Is Johnston using GPS only? Not too clear from the paper.

GNSS.

Figure 7: (1) there are several L2 retrieval symbols with no SNR symbol next to it. Is this really a fair comparison? What happens if only collocated data is used?; (2) I assume these are g/kg units shown?; (3) these "bigger symbols", are those the vertical lines that are sometimes visible, sometimes not? Please improve readability here. (4) again, consider adding some further info, e.g. number of data points, overall correlation stats, e.g. bias, std dev, correlation coefficient.

Do you mean some orange samples from ARRecon_2020 campaign? The fact that filled orange triangles (SNR-ML retrievals) are closer to the 1:1 line than open triangles (RO product) indicates that SNR-ML method works better for the atmospheric river scenario in the MABL. Some other studies used the ARRecon, GNSS-RO and ERA5 collocation data also reported refractivity negative bias in GNSS-RO (COSMIC-2, SPIRES) and dry bias in ERA5 in MABL (Murphy and Haase, 2022).

Murphy, M. J., J. S. Haase: Evaluation of GNSS Radio Occultation Profiles in the Vicinity of Atmospheric Rivers, Atmosphere, **https://doi.org/10.3390/atmos13091495**.

We now modified Fig. 7 (now Fig. 9) to incorporate suggestions from both reviewers. Please note that both Fig. 9 and Fig. 10 are provided in huge image size with very high dpi resolution considering the rich information that has to be contained in these two figures. Interested readers can enlarge the plots to check more details.

Figure 9: I am unsure if this comparison is fair towards ERA5! There are e.g. 549 sondes for Magic (Figure 3), but about 10% or less are shown for L2 and SNR, while I guess ERA5 shows the correlations for all. Same with the other campaigns, or rather even worse, EUREC4A has 1349 sondes, but only 53 are shown for SNR.

We are sorry for the misunderstanding here. We had claimed in the Fig. 9 (now Fig. 11) caption that the correlations were made for the sub-samples for ERA-5 where a SNR-radiosonde collocation is identified, and that's why we didn't write the number of samples above ERA-5 violins. It is indeed very hard to find a radiosonde-GNSS collocation. Even though we added Metop-A and Metop-B predictions in this revised version, it only added two more collocation samples among all 6 campaigns.

Most importantly, we identified two bugs in our plotting codes for Fig. 9 (now Fig. 11). For ERA-5, we didn't filter out missing values when compute the correlation, causing the correlation coefficients to be very small. We have fixed the bug and now all three datasets look quite comparable in the updated Fig. 11. We also found that the success rate for both SNR-ML and wetPrf methods are pressure-level dependent. Hence, we now provide a new Table 3 to illustrate the robust success rate using the SNR-ML method. The entire Section 3.3 has been largely rewritten with the new updated results.

P16/L256: What is the reason to exclude sea ice (above it talks of AIRS issues)? The MERRA model? Or the L2 retrieval? The SNR one?

Because we worry sea ice induced reflectometry signal could contaminate our deepSNR based retrievals.

Figure 10: Any reason why these plots do not cover the same respective area? Are you comparing the same numbers/locations/times here? Is the SNR based retrieval maybe picking primarily wet conditions and having trouble with dryer situations, thus leading to this higher humidity? How is topography taken care of, e.g. for Norway in SNR retrievals?

Because we used 2.5 X 2.5 degree resolution for MERRA-2, but have to use 4 X 4 degree resolution due to the sparsity of GNSS data during the boreal winter season.

P17L258-262: I think these last sentences would need much further assessment, e.g., you mention topography limitations when discussing Figure 5, I am still not sure whether you are comparing like with like, the SNR retrieval has several quality controls implemented (that seem to filter out dryer observations, e.g. discussion with Figure 6), etc.

Are you saying that because we filtered out weak SNR profiles, our retrievals tend to biased toward wet-conditions? If that is the case, how to explain the dry-bias in the southern ocean for those very small values? Another evidence that this SNR-ML retrieval product is not inherently wet-biased is the comparison between Fig. 10 and Fig. 11 (now Fig. 12 and Fig. 13). In the Antarctic region, MERRA-2 doesn't show systematic dry bias.

P17/L265: Again, topography might have an impact on your sampling, also the dry quality control filtering, etc. Thus SNR needs to show that it is doing a better job against an independent data set. Re-analysis models might just also get it wrong there (or the sampling is not consistent).

True. I agree with you.

Figure 12: Again, is this comparing the same data at the same location and time? Or just taking all available SNR retrievals in the area, and comparing it to all ERA5 data in that box?

We took all available retrievals during Dec.-Feb. period listed in Table 1 in a given region to construct the diurnal cycle. Since these campaigns were all carried in the pre-COSMIC era, the campaign data cannot be used to verify

Table A1: What is the ascending / descending orbit? Is that of any relevance? Or is this the occultation setting / rising?

You are correct. It's occultation setting and rising. For typical satellite community users, ascending/descending is a more familiar terminology, but we had added the clarification in the Table caption now. The caption now reads:

Table A1. Summary of GNSS-RO instrument noise (σ) used in this work. "Ascending" and "descending" here are equivalent meaning of occultation rising and setting.

Figure A1: Why are you using different labels and titles here, compared to the COSMIC one you show? And is this really Metop, as the caption talks of COSMIC? And why is this the only Metop result shown?

We have reedited Fig.2, Fig. A1 and Fig. A2 to show the correlation situation for COSMIC-1, METOP-A and METOP-B for SNR and sigma^2_SNR. The reasons to show the three is to demonstrate that (1) the correlation is not highly non-linear, so a ML model is better to handle this rather than a single-level regression or multi-variable regression model; (2) the correlation pattern is highly instrument dependent, so we need to build individual ML models for each mission separately.

I noted that the other reviewer was very thorough also regarding typos, textual improvements, thus I mostly did not include those here. But a thorough re-editing is needed!

Thanks again for your constructive suggestions. If you have time, please feel free to also read the responses to reviewer #1's questions. With compiling the suggestions from you two, we hope the manuscript now is improved for publication. Thanks!

---

## Author Response (AR2)

**Public justification (visible to the public if the article is accepted and published)**:
I appreciate the revised version of the manuscript; I am, in particular, grateful to the authors for pointing out their mistake in calculating some results that, in my view, change the conclusions reported in the paper significantly. The discussions on the ML methodologies applied and the physical background of the relation between PBL humidity and SNR measurements are important additions that make the paper considerably more useful for its audience.

Dear editor,

Thank you sincerely for reading through the revised manuscript and the response, and provide your suggestions. We have now modified Fig. 9 and 10 (formerly Fig. 7 and 8) according to both reviewers and your suggestions (see response #1 below). We add more in-depth discussions regarding each campaign/weather regime (see response #2 below). For Fig. 11, the extrapolated non-meaningful values beyond +1 and -1 due to python sns.violinplot artifacts have now been fixed. Please see response #3 for refinements related to the discussion of Fig. 11.

Nevertheless, I am convinced that the manuscript still needs major revisions, for the following reasons:

1) Both reviewers suggest changes to Figures 9 (formerly 7) and 10 (formerly 8); reviewer 1 made more specific suggestions to reduce the number of levels, enlarge the figure size, and maybe reduce the number of levels being shown. I fully agree with their assessment, even for the redrawn figures. For example, the ERA5 line is still not visible, nor are the mean values for each campaign visible; the figures are too busy. Increasing the size on a computer screen as suggested by the authors does not solve the problem. I can see no qualitative difference between the three figures for the 975, 950 and 925 hPa levels, nor between the 900 and 875 hPa levels.

Response #1: Following the suggestions by reviewers and the editor, we now doubling the line thickness of ERA5 result in Fig. 9 (gray thick solid line) and put it underneath the SNR result (black thick solid line), so readers can see the SNR-ML results track ERA5 closely, often overlapping each other, meaning that the SNR retrieval is comparable in quality with ERA5 reanalysis. The mean of each campaign is now not only plotted in a bigger symbol, but also encircled by a black boundary to make sure the mean and standard deviation of each campaign stands out from individual samples in both Fig. 9

and Fig. 10. The subset of ERA5 collocation samples to generate the gray line in Fig. 9 is now plotted as open symbols in Fig. 10 to better visual comparison. The black thick solid line connecting each campaign's mean in Fig. 10 is enhanced in thickness now.

We still believe it's important to show individual comparison samples in the background to demonstrate the variation and extremes in one campaign that standard deviation is not good enough to capture. For example, in Fig. 9 SNR retrievals during ARRecon_2020 (filled orange triangles) apparently outperform L2 (open orange triangles) for the extremely wet situations and relatively dry situations, even though their means are indifferentiable for this campaign at 900 – 850 hPa. In Fig. 10, we can see although the means from all 6 campaigns track the 1:1 line closely with slight dry-bias, there are many much-too-wet values in ERA5 during the EUREC4A campaign (blue dots that are away from the 1:1 line) in the lower levels. Also, although the mean comparison during the MARCUS campaign (cyan triangles) seems to show that ERA5 does a good job in the polar PBL, one can see the collocation samples deviate from the 1:1 line when mixing ratio is smaller than 3 g/kg.

In the text, the authors also do not discuss individual levels in detail; instead, they make qualitative statements such as "In general, better correlation are found when…" without pointing to any evidence in the figure. In the text, the authors also discuss only results from the ARRecon campaign in detail. The figure is clearly important, but the many details are neither discussed, nor can they be easily deducted from the figure alone. The authors may also consider other ways of representing their results, e.g. showing data for individual campaigns, as in Fig. 11
Response #2: Now the discussions around Fig. 9 and Fig. 10 have been revised substantially to add in more in-depth discussions and comparisons. It worths clarifying that Fig. 11 is the only figure that we identified a bug in coding, which changed some conclusions related to Fig. 11, but not to Fig. 9 nor Fig. 10.

[revised manuscript text omitted]

2) The correction of the calculation of correlation coefficients has changed Figure 11 considerably. I agree with the authors that the performance of all three data sets is

miserable for the tropical campaigns (EUREC4A and ATOMIC); and SNR-ML outperforms ERA5 and wetPrf/wetPf2 for ARRecon. However, the authors should also point out that ERA5 outperforms both SNR-ML and wetPrf/wetPf2 during the MARCUS and MAGIC campaigns, and that wetPrf/wetPf2 still outperforms SNR-ML during MARCUS. Thus, (page 19, line 331 onwards) it is not correct that "the quality of SNR-ML retrievals is comparable to ERA 5 and operational wetPrf/wetPf2 products", and that it even outperforms the other two in the ARRecon case. At best, the result is inconclusive; SNR-ML outperforms the other two in one case, underperforms in two cases, and is as useless as the others in another two cases. Also, the statement (page 24, line 425 onwards) that the results demonstrate "the real information content in the SNR signal is learnt" during the ARRecon campaign should be extended to say that in two other campaigns, the SNR-ML retrieval failed to extract this information from the data. A similar update is required in the abstract (page 1, lines 13 - 16); the SNR-ML also underperforms compared to ERA5, and the ML retrieval did not extract useful information from the SNR signal in these cases.

There is more to say on Fig. 11 b: Why do the violin plots indicate the presence of correlations with values > 1? The box plots inside the density estimates indicate no such data, but maybe a violin plot then is not a good way to show the characteristics of the data. Or the artefact should be mentioned in the discussion.

Response #3: The values beyond +1 and -1 originate from missing use a cutoff parameter in using the python sns.violinplot function. Now the cutoff has been added to cut out any artifacts beyond maximum and minimal values.

We partially agree with your interpretations above and had incorporated some into the discussions. Thank you very much. Specifically, we agree that "the performance of all three datasets is miserable for the tropical campaigns (EUREC4A and ATOMIC); and SNR-ML outperforms ERA5 and wetPrf/wetPf2 for ARRecon." However, we believe your suggestion that "ERA5 outperforms both SNR-ML and wetPrf/wetPf2 during the MARCUS and MAGIC campaigns, and that wetPrf/wetPf2 still outperforms SNR-ML during MARCUS" is not 100% accurate. For the MARCUS campaign, ERA-5 is apparently dry-biased when specific humidity < 3g/kg (now mentioned the in the discussion of Fig. 10 in the revised manuscript). This bias seems to be slightly mitigated using the SNR-ML method. If you revisit Fig. 6, SNR-ML method produces much larger variations for very dry situations than ERA-5, which however also comes with large uncertainty to make it a robust retrieval. Moreover, we recently learnt from radiosonde payload

provider that the humidity sensor response time on radiosonde is significantly delayed at extremely low temperature (usually happens around tropopause, but could also happen over polar winters), in which case the radiosonde readings might not be trustworthy as the "ground truth". So we agree with your suggestion that results for MARCUS campaign (i.e., Southern ocean) is inconclusive. We have now included that in the discussion of Fig. 11.

For the MAGIC campaign, SNR-ML method actually outperforms the wetPrf/wetPf2 products in both the correlation coefficients (Fig. 11a) as well as the number of available samples especially at the lowest three pressure levels (Table 3). As super-refraction tends to occur more frequently at stratocumulus regions (e.g., Xie et al., 2010), SNR-ML method possesses unique advantage over wetPrf/wetPf2 products in providing unbiased PBL humidity retrievals.

Regarding editor's concern about violin plots, after fixing the artifacts, we believe this is our best plotting option to integrate multi-dimensional information into one figure. The skewness of the distribution of correlation coefficients, previously not discussed, is now included in the discussion as well. Fig. 11b in particular demonstrates the robustness of the SNR-ML retrievals across all 6 PBL pressure levels: although the highest positive correlations are always identified in ERA-5 and/or wetPrf/wetPf2 products, the medians of SNR-ML retrievals are consistently the highest with consistent top-heavy distribution except for 850 hPa, meaning that SNR-ML retrievals agree with radiosonde "truth" more consistently while ERA-5 and wetPrf/wetPf2 have more variations. Of course all these conclusions are limited by the smaller collocation samples (<=309 in total), and we for sure need more extensive evaluation for this research product before massive production.

The violin plots in Fig. 11 and number of collocated sample statistics in Table 3 help disentangle the merits/caveats of SNR-ML retrievals from multi-dimensional statistical metrics.  Only correlation coefficients of all collocated samples collected from each campaign are displayed in Fig. 11. The ARRecon-2018 and ARRecon-2020 samples are further combined. From Fig. 11a, we can see both  and Level 2 retrieval  again that the MABL specific humidity is not well captured in the tropics by either of the three datasets (EUREC4A and ATOMIC), but SNR-ML

generated retrievals perform slightly better than the operational wetPf2 products in the deep tropics and trade-cumulus regions. In the rest three campaigns in the mid- and high-latitudes, they all agree very well with the radiosonde/dropsonde ground truths. ERA-5 reanalysis does the best job at high-latitude southern ocean (MARCUS) as well as the stratocumulus region (MAGIC), while  in the atmospheric river regime, SNR-ML retrievals outperform the wetPrf/wetPf2 retrievals as well as the ERA-5 reanalysis. It is worth noting that SNR-ML retrievals perform slightly better than wetPrf/wetPf2 retrievals in the stratocumulus region (MAGIC) in both the medians and the top-heavy skewness of its distributions, which can be partially attributed to the scarcity of wetPrf/wetPf2 collocation samples in this weather regime and known bias in the Level 2 retrieved refractivity gradient (Xie et al. (2010)). For the polar region (MARCUS), although SNR-ML retrievals exhibit the lowest correlations among the three datasets albeit all correlations are statistically significant, it is inclusive at this point to say that SNR-ML method is not suitable for the polar region. As a matter of fact, SNR-ML method generates the largest variabilities among the three when the PBL is extremely dry (Fig. 6), but the SNR in this situation is generally too weak to generate a robust retrieval (i.e., uncertainty too large compared to retrieved value). The retrievals from the SNR-ML method at dry polar winters contain more potentials (e.g., Fig. 14 as an example) if future GNSS missions could improve the SNR.

~~The SNR-ML retrieval exhibits more robust correlation while Level 2 retrieval are really poor (negative correlations) at some levels. However, for the two deep-tropics campaigns, above statements do not work anymore. All three work really poor for the ATOMIC campaign with barely any correlation with the ground truth or even negative correlation for the Level 2 retrieval. For the EUREC4A campaign, things remain similar for RO retrievals no matter using Level 1 or Level 2 data, but ERA-5 works much better at capturing the humid MABL structure in this case albeit it's still dry-biased (not shown). We achieved 20-80% across-board more collocation samples using the SNR-ML retrievals versus the Level 2 wetPrf or wetPf2 retrievals, the latter of which is consistent with the general success rate shown in Fig. 1.~~

Fig. 11b demonstrates the robustness of the SNR-ML retrievals across all 6 PBL pressure levels. Although the highest positive correlations are always identified in ERA-5 and/or wetPrf/wetPf2 products, the medians of SNR-ML retrievals are consistently the highest with consistent top-heavy distribution except for 850 hPa, meaning that SNR-ML retrievals agree with radiosonde/dropsonde "truths" more consistently while ERA-5 and wetPrf/wetPf2 have more variations across different weather regimes. Of course all these conclusions are limited by the smaller collocation samples (<=309 in total), and we for sure need more extensive evaluation for this research product before massive production.

3) Neither reviewer discussed the ML approach in detail. If I understand correctly, the authors train the data on one period, perform the hyperparameter tuning on a test period, but then evaluate (or validate) the fitted model on the combine training and test data. Isn't this a case of data leakage?

You are correct. There is a data leakage in the prediction period, but the hyperparameter tuning strictly follows the standard ML procedure. The data leakage only affects the COSMIC-1 data and the MAGIC campaign comparison. We unfortunately didn't have enough disk array and computational powers at the time when performing the training/tuning. The reason for including both training and testing data for prediction is for generating robust enough samples for constructing statistically meaning climatology, especially the diurnal cycle (COSMIC-2 couldn't cover high-latitude). We now mentioned this issue in the revised manuscript.

In this work, we created a collocated and coincident ERA-5 - SNR training and validation dataset. The SNR records are from  four satellite series: COSMIC-1, COSMIC-2 METOP-A and METOP-B. The periods for training, independent testing, and prediction are listed in Table 1. Note that the  testing period is independent from training period to avoid potential self-correlation using standard random splitting procedure. The prediction period however covers both training and validation periods mainly for generating enough samples to construct statistically robust climatology (e.g., diurnal cycles). This however creates an unfortunate data leakage concern for the comparison with the MAGIC campaign but not for the rest of other independent validation datasets (Table 2). The target variables are specific humidity at the aforementioned 6 pressure levels ($975hPa$, $950hPa$, $925hPa$, $900hPa$, $875hPa$ and $850hPa$). The input parameters are 52 levels of $S_{RO}$, 52 levels of $\sigma_S^2$, latitude, longitude, month and Rising/Setting flag.

In addition, the authors mention results from fitting other ML algorithms, such as gradient-boosted trees; it would be good to give references. The authors state that results obtained with these ML models were comparable with theirs; were hyperparameters also tuned, or would that provide an even better performance? I was also confused by "logistic regression" and "Support Vector Machines (SVM)" being mentioned. After all, they are classification algorithms, so why should they be applied here?

SVM, random forest and gradient boosting methods tested in this paper are all employed from the standard "scikit-learn" package. See https://scikit-learn.org/stable/modules/generated/sklearn.svm.SVR.html for the SVM regressor (should be called SVR indeed). We did minimal hyperparameter tuning for these methods, mainly just to check the performance metrics and the important factor ranking to assure physical consistency (see response letter Page 23). These are shallow ML models that are considered "outdated" in ML field these days, so we didn't explore further of these models. I had a wrong memory about linear regression, which I never used. This has been crossed out in the revised version.

MLP regressor is employed from the pytorch library. We performed hyperparameter tuning for this one and the performance is slightly worse but overall quite comparable to CNN. There is really no preference of a ML model for this paper. We stated the reason why we didn't perform extensive tuning and cross-ML model comparisons as this is not the focus of this paper. See below revised text:

We also tried some earlier old-fashioned ML models, e.g., random forest (RF), gradient boosting (GB), logistic regression (LR), support vector machine (SVM) and one deep learning model multilayer perceptron (MLP). The model performances are actually very close in terms of evaluating the RMSE except for the LR and SVM, the latter of which performed discernibly worse than the rest ML models. It is not a surprise finding as this is a relatively simple and straightforward task that ML models should handle easily, but not the case for multi-variable linear regression type of logistic models (hence, it explains the poor performance of LR and SVM). As the main focus of this paper is science and new information content embedded in SNR signals, we will not deviate the attention to spend more time discussing these model results. The semi-transparency of RF and GB models is appreciated by us though. We compared the feature importance rankings with Wu et al. (2022) findings, and find high consistencies (e.g., high ranking of SNR at $H_{SL} = -100\ km$ in the tropics, and SNR at $H_{SL} = -80\ km$ ranks the top in the polar region).

4) Table 1 (page 6) states that prediction intervals for Metop-A and -B were in "2012.01 - 2011.12". Apart from the wrong order of the orders, this is strange given that Metop-B was launched in late 2012 only. Please review the entries in this table carefully.

Thanks for identifying this typo. It should be 2012.01-2012.12. Typos are fixed now.

5) I believe Table B1 also contains wrong data. Excess phases increase towards the ground. Thus, increasing values of log(excess phase) correspond to data lower down in the atmosphere, and hence Hsl. However, the table claims that the largest excess values correspond to the highest Hsl.

Thanks for pointing that out! I went back checking the corresponding value and indeed the log(excess phase) should be reversed.

Finally, I strongly recommend to have a native speaker review the text before the resubmission of the manuscript. There are various leftovers from LaTeX code (e.g., "textcolorred" on page 7 line 160) and incomplete sentences that require a thorough review.

Thank you! We have scrutinized the cleaned version to make sure there are no leftover LaTex codes as well as no grammar errors (to the best we can).

---

## Author Response (AR3)

Thank you for scrutinizing the revised manuscript, our responses and provide your constructive suggestions from the original version all the way through. Please see our refinements/response below in blue letters. We also proofread the revised manuscript and corrected several grammar errors.

**Public justification (visible to the public if the article is accepted and published)**: Thank you very much for the updated version of your paper.

You have addressed nearly all of my concerns; in particular, the figures 9 and 10 are finally drawn in a way that your argumentation in the text can be followed, and the latter has alse greatly improved. I am also grateful for you stating the limitations and shortcomings of the work described in the paper. With that, the paper is nearly ready for publication, with the exception of the following details:

1. I mentioned in my previous communication that Metop-B was launched in late 2012; you can find the precise date by googling. I thus do not understand why even in your corrected version, you claim you used (non-existent) data from that satellite during 20212. Please correct the entry in Table 1 by checking which data you really used. While the correction will not change your results and conclusion, it may create doubts on your thoroughness and integrity for some readers; don't give them such an argument. Thank you so much for reiterating this point, which I apparently overlooked in the previous response. You are absolutely correct. After checking back (see below), our SNR data are indeed only available since 2014.032 for Metop-b, and 2007.274 for Metop-a. This mistake has been corrected in Table 1. In addition, the inconsistent writings of "Metop" and "METOP" have now been all changed to "Metop".

```
(3point6) [jgong@gs613-cirrus atmPhs]$ pwd
/data11/dlwu/gps/phs/metopb2016/atmPhs
(3point6) [jgong@gs613-cirrus atmPhs]$ ls -l
total 0
drwxr-xr-x. 1 dlwu dlwu 65110 Dec  3  2021 2013.032
drwxr-xr-x. 1 dlwu dlwu 62674 Dec  3  2021 2013.033
drwxr-xr-x. 1 dlwu dlwu 62870 Dec  3  2021 2013.034
drwxr-xr-x. 1 dlwu dlwu 61288 Apr 10  2021 2013.035
drwxr-xr-x. 1 dlwu dlwu 60818 Apr 10  2021 2013.036
drwxr-xr-x. 1 dlwu dlwu 61092 Dec  3  2021 2013.037
drwxr-xr-x. 1 dlwu dlwu 60724 Apr 10  2021 2013.038
drwxr-xr-x. 1 dlwu dlwu 62032 Dec  3  2021 2013.039
drwxr-xr-x. 1 dlwu dlwu 61852 Apr 10  2021 2013.040
drwxr-xr-x. 1 dlwu dlwu 59674 Dec  3  2021 2013.041
drwxr-xr-x. 1 dlwu dlwu 61382 Apr 10  2021 2013.042
drwxr-xr-x. 1 dlwu dlwu 60442 Apr 10  2021 2013.043
drwxr-xr-x. 1 dlwu dlwu 61264 Dec  2  2021 2013.044
drwxr-xr-x. 1 dlwu dlwu 61280 Dec  3  2021 2013.045
drwxr-xr-x. 1 dlwu dlwu 57144 Dec  2  2021 2013.046
drwxr-xr-x. 1 dlwu dlwu 60912 Apr 10  2021 2013.047
drwxr-xr-x. 1 dlwu dlwu 61938 Dec  2  2021 2013.048
drwxr-xr-x. 1 dlwu dlwu 56956 Dec  3  2021 2013.049
drwxr-xr-x. 1 dlwu dlwu 60896 Dec  3  2021 2013.050
drwxr-xr-x. 1 dlwu dlwu 59588 Dec  3  2021 2013.051
drwxr-xr-x. 1 dlwu dlwu 64828 Dec  3  2021 2013.052
drwxr-xr-x. 1 dlwu dlwu 61288 Apr 10  2021 2013.053
```

```
(3point6) [jgong@gs613-cirrus atmPhs]$ pwd
/data11/dlwu/gps/phs/metopa2016/atmPhs
(3point6) [jgong@gs613-cirrus atmPhs]$ ls -l
total 0
drwxr-xr-x. 1 dlwu dlwu 64578 Dec  4  2021 2007.274
drwxr-xr-x. 1 dlwu dlwu 63356 Dec  3  2021 2007.275
drwxr-xr-x. 1 dlwu dlwu 64484 Dec  3  2021 2007.276
drwxr-xr-x. 1 dlwu dlwu 64954 Dec  3  2021 2007.277
drwxr-xr-x. 1 dlwu dlwu 61476 Dec  3  2021 2007.278
drwxr-xr-x. 1 dlwu dlwu 64860 Dec  3  2021 2007.279
drwxr-xr-x. 1 dlwu dlwu 64954 Dec  3  2021 2007.280
drwxr-xr-x. 1 dlwu dlwu 62510 Dec  2  2021 2007.281
drwxr-xr-x. 1 dlwu dlwu 62792 Dec  3  2021 2007.282
drwxr-xr-x. 1 dlwu dlwu 64672 Dec  3  2021 2007.283
drwxr-xr-x. 1 dlwu dlwu 64014 Dec  2  2021 2007.284
drwxr-xr-x. 1 dlwu dlwu 65706 Dec  3  2021 2007.285
drwxr-xr-x. 1 dlwu dlwu 64296 Dec  2  2021 2007.286
drwxr-xr-x. 1 dlwu dlwu 64954 Dec  3  2021 2007.287
drwxr-xr-x. 1 dlwu dlwu 56118 Dec  2  2021 2007.288
drwxr-xr-x. 1 dlwu dlwu 65236 Dec  3  2021 2007.289
drwxr-xr-x. 1 dlwu dlwu 66458 Dec  3  2021 2007.290
drwxr-xr-x. 1 dlwu dlwu 64578 Dec  3  2021 2007.291
drwxr-xr-x. 1 dlwu dlwu 64202 Dec  3  2021 2007.292
drwxr-xr-x. 1 dlwu dlwu 65518 Dec  3  2021 2007.293
drwxr-xr-x. 1 dlwu dlwu 65048 Dec  2  2021 2007.294
drwxr-xr-x. 1 dlwu dlwu 66270 Dec  2  2021 2007.295
drwxr-xr-x. 1 dlwu dlwu 63826 Dec  2  2021 2007.296
```

Fig. R1: screenshots of SNR data that we processed for Metop-b (left) and Metop-a (right). We started processing Metop-b data since 2013.032 and Metop-a data since 2007.274.

2. Editorial: Line 326: "Firstly, The quality of..." -> "Firstly, the quality of..." (turn the capital letter 'T' into lower case).
Corrected. Thanks.

With these changes, the paper can be published.

Allow me a few additional remarks on your paper; I do not expect you to include a discussion on these points in the final version.

Grinsztajn et al. (2022) (also see references therein) demonstrated that the "old-fashioned" Statistical Learning methods like Gradient Boosted Trees regularly outperform deep learning approaches on tabular data. Only very recently, Hollmann et al. (2025) published a transformer-based neural network that seems to overcome this problem. As you are using a much simpler neural network design than the one proposed in the latter paper, it might well be that your desire to be more fashionable and follow the hype on deep learning methods made you miss better results you might have been able to achieve simply by performing hyperparameter tuning. That is a pity.

Thanks for recommending this paper. While I do not have bandwidth to read this paper closely right now, I agree totally with you that newer ML methods not necessarily work better than old ML methods. In this paper, we put some efforts in the early stage of this work for hyperparameter tuning with random forest and gradient boosting. As mentioned in the current manuscript, the best performance in terms of minimizing RMSE is quite comparable across simpler ML models and deep-learning models, and we chose CNN as our final model mainly because it learns the cross-correlation between different pressure layers, not because it outperforms other simpler ML models. For your interests, we do have ongoing works adopting transformer-based or diffusion-based models into Earth science application domains, which simple MLs cannot be used anymore (e.g.,

https://doi.org/10.48550/arXiv.2411.17000 ). However, due to funding cycle limitation, we cannot extend efforts into adopting these new ML models into this specific

Data leakage has been identified as a significant issue in ML-based scientific applications; see Kapoor et al. (2023), who even refer to a "reproducibility crisis" in machine-learning-based science. The workshop page at https://sites.google.com/princeton.edu/rep-workshop/ offers additional insights, particularly through its annotated reading list. That is why it is essential to avoid data leakage, or at least discuss it if it has occurred (as you are doing now).

Thank you very much for bringing this paper to our attention. We have added it as an addition citation in the updated manuscript.

References:

Grinsztajn, Leo, Edouard Oyallon, and Gael Varoquaux. 'Why Do Tree-Based Models Still Outperform Deep
Learning on Typical Tabular Data?' Advances in Neural Information Processing Systems 35
(6 December 2022): 507–20.
https://proceedings.neurips.cc/paper_files/paper/2022/hash/0378c7692da36807bdec87ab043cdadc-Abstract-Datasets_and_Benchmarks.html

Hollmann, Noah, Samuel Müller, Lennart Purucker, Arjun Krishnakumar, Max Körfer, Shi Bin Hoo,
Robin Tibor Schirrmeister, and Frank Hutter. 'Accurate Predictions on Small Data with a Tabular
Foundation Model'. Nature 637, no. 8045 (January 2025): 319–26.
https://doi.org/10.1038/s41586-024-08328-6.

Kapoor, Sayash, and Arvind Narayanan. 'Leakage and the Reproducibility Crisis in Machine-Learning-Based
Science'. Patterns 4, no. 9 (8 September 2023).
https://doi.org/10.1016/j.patter.2023.100804.